# Effect of Reactive Oxygen Species Photoproduced in Different Water Matrices on the Photostability of Gadusolate and Mycosporine-Serinol

**DOI:** 10.3390/md22100473

**Published:** 2024-10-16

**Authors:** Martin George Thomas, Sylvie Blanc, Mickael Le Bechec, Thierry Pigot, Susana C. M. Fernandes

**Affiliations:** IPREM—Institute of Analytical Sciences and Physico-Chemistry for Environment and Materials, Universite de Pau et des Pays de l’Adour, E2S UPPA, CNRS, 64000 Pau, France; martinpsrt@gmail.com (M.G.T.); sylvie.blanc@univ-pau.fr (S.B.); mickael.lebechec@univ-pau.fr (M.L.B.)

**Keywords:** mycosporines, gadusol, UV filters, photostability, photodegradation, photosensitizer, porphine, riboflavin, aquatic matrices

## Abstract

In the past few years, there has been an increasing interest in mycosporines—UV-absorbing molecules—bringing important insights into their intrinsic properties as natural sunscreens. Herein, mycosporine-serinol and gadusol (enolate form)/gadusolate were exposed to UV radiation via a solar simulator and the photostability was assessed in pure water and different natural matrices like river, estuary and ocean water. In general, this study revealed that the photodegradation of gadusolate and mycosporine-serinol was higher in natural matrices than in pure water due to the generation of singlet oxygen on UV irradiation. In pure water, in terms of photostability, both gadusolate and mycosporine-serinol were found to offer good protection and high performance in terms of photodegradation quantum yield ((0.8 ± 0.2) × 10^−4^ and (1.1 ± 0.6) × 10^−4^, respectively). Nonetheless, the photostability of mycosporine-serinol was found to be superior to that of gadusolate in natural water, namely, ocean, estuary and river. The present work highlights how mycosporine-serinol and gadusolate resist photodegradation, and supports their role as effective and stable UV-B sunscreens.

## 1. Introduction

Among the natural molecules acting as ultraviolet (UV) sunscreens, mycosporines and mycosporine-like amino acids (MAAs) are some of the most prominent exemplars [1,2,3]. Mycosporines are UV-absorbing compounds at around 310 nm which possess a cyclohexenone ring system linked with an amino acid or amino alcohol. On the other hand, MAAs are UV-absorbing compounds with imine derivatives of mycosporines or enamino imines that consist of a cyclohexenimine ring system with UV absorption maxima between 310 and 360 nm. Both have high molar extinction coefficients (from 28,100 to 60,000 L·mol^−1^·cm^−1^), the ability to dissipate the absorbed UV radiation as heat energy without generation of oxidative photo-products, and high photo- and thermostability [1,2,3,4,5,6,7,8,9]. These two families of secondary metabolites are biosynthesized by algae, fungi and cyanobacteria, and ingested or accumulated by marine animals like fish, mollusks and arthropods to protect themselves from the harmful effects of UV radiation [1,2,4]. Mycosporines and MAAs are low molecular weight (<400 Da) and water-soluble molecules. Their precursors, gadusol and 6-deoxygadusol, consist of cyclohexenone compounds absorbing in the UV-B spectrum [3,7,10].

Because of some controversy related to side effects provoked by conventional synthetic sunscreens and mineral sunscreens (e.g., TiO_2_, ZnO, etc.) on both human and environmental health, and their limited photostability and cross-stability with other sunscreen agents, these natural sunscreens have attracted the interest of the academic community, scientists and industrials [10,11]. Due to widespread use of synthetic sunscreens along with recreational activities, these compounds end up in aquatic ecosystems like oceans, rivers and estuaries [12]. Consequently, this can cause neurotoxicity in marine organisms, increase the mortality in fish, reduce coral reproduction and exacerbate coral reef bleaching [13,14]. Thus, these UV-absorbing natural molecules are suitable potential candidates as an alternative to conventional synthetic UV filters in sunscreen formulations [4,15] and the design of UV-absorbing and protective biomaterials [16]. Indeed, the assessment and understanding of their photostability in natural water matrices and in the presence of photosensitizers seems to be crucial. Herein, there is a particular focus on natural molecules absorbing in UV-B (Figure 1). The penetration of UV-B rays into the Earth’s atmosphere can cause a series of side effects in humans such as sunburn, immunosuppression and generation of reactive oxygen species (ROS), leading to DNA damage and carcinogenesis (melanoma and non-melanoma skin cancer) [7,16].

The organic matter (e.g., humic acid, fulvic acid) present in natural water matrices, like river, estuary and ocean, is known to generate ROS upon solar irradiation and thus helps to transform organic molecules, whether natural or made by humans. To highlight the role of ROS in the abiotic degradation process, a detailed study on the photostability of mycosporine-serinol and MAAs was previously done, and the authors concluded that these molecules are very stable in terms of pH and temperature [20].

In this context, we propose to investigate the photostability of mycosporine-serinol and the enolate form of gadusol (gadusolate) under realistic conditions, i.e., under solar irradiation and in natural waters (river, estuary and ocean). Complementary experiments conducted by introducing photosensitizers—riboflavin and porphine—into pure water to simulate ROS production are also proposed, as well as the use of pure water, mineral water and artificial seawater for comparative purposes. Only two studies have investigated the photostability of gadusol and its derivatives. Arbeloa et al. [21] studied the photostability of gadusol in aqueous solution at neutral pH 7 (gadusolate-enolate form with λ_max_ = 296 nm) and acidic pH 2.5 (gadusol-enol form with λ_max_ = 268 nm) under air or argon atmospheric conditions. Under physiological conditions, they demonstrated the high photostability of gadusol/gadusolate even if gadusolate presented lower quantum yields of photodegradation (1.38 ± 0.45 × 10^−4^) than gadusol (3.64 ± 0.36 × 10^−2^). With gadusolate, the authors concluded that the reductive quenching reactivity (electron transfer from gadusolate to excited triplet state sensitizer Type 1 mechanism) could be considered as one of the fundamental mechanisms that support the antioxidant capacity of gadusol in biological environments. Interestingly, the same authors also demonstrated that gadusolate reacted very efficiently with singlet oxygen arising from energy transfer from an excited state sensitizer (Type 2 mechanism), which also contributes to the antioxidant properties of gadusolate [22].

To the best of our knowledge, the photolysis of mycosporine-serinol in the presence of flavin was briefly studied by Bernillon et al. [23], but its photostability has never been reported before. In this study, the photostability of gadsusolate and mycosporine-serinol is thoroughly investigated in different natural matrices and in the presence of photosensitizers.

## 2. Results and Discussion

Before studying the stability of M-ser(OH) and gadusolate, the different waters were characterized (pH, absorbance, light reactivity) in order to evaluate the different matrix parameters involved in the photodegradation mechanisms.

### 2.1. Properties of Matrices

The pH, which affects most chemical and biological processes in the water and the form (acid or basic) of M-ser(OH) and gadusolate, was measured for all matrices. It was equal to 5.3 in pure water, 7.2 in mineral water and slightly basic for the ocean, river and estuary water, with a pH of 8.1, 8.4 and 8.5, respectively. The pH of artificial seawater was fixed at 8.2. These values are in agreement with those reported on water from various sources by Kulthanan et al. [24]. Natural waters contain a number of weak acids and bases that control the pH, and the most abundant of these species are carbonate and bicarbonate ions [25]. Changing the proportion of these species in equilibrium induces a pH variation [26]. A higher concentration of carbonate increases the pH, as is probably the case in estuarine waters, and a lower concentration leads to a decrease in pH, as is observed for seawater. The absorption spectra of the different waters were measured from 250 to 700 nm at room temperature; they are presented in Figure 2. As expected, pure and mineral waters and artificial seawater do not have significant absorbance. However, an absorption was observed for river and estuary waters with a spectrum similar to that recorded in pure water for humic substances [27] present in natural matrices. The low absorption measured for ocean water has already been reported with absorbance values very close to those of surface waters in the tropical Atlantic Ocean [28]. According to the literature [29], the absorbance intensity is correlated with the concentration of colored dissolved organic matter (CDOM), chemically complex organic compounds which can absorb UV-A, UV-B and visible light. Estuary water, resulting from the transfer of water from the river to seawater, has an absorption value (A_250nm_ = 0.18) between that of the river (A_250nm_ = 0.45) and that of the ocean (A_250nm_ = 0.05) leading to a dilution factor of around 3 for CDOM relative to the CDOM in the river.

The colored dissolved organic matter can act as a natural photosensitizer generating, in the presence of sunlight, reactive intermediates such as reactive oxygen species (ROS) [30] by energy or electron transfer from its excited state to molecular oxygen in surface waters. Among the ROS, singlet oxygen ^1^O_2_ is recognized as a potentially important species of indirect photolysis of organic compounds in natural water [31]. Its steady state concentration ([^1^O_2_]ss) was evaluated in river, ocean and estuary water using the molecular probe furfuryl alcohol (FFA) chosen due to its high selectivity and a high second-order reaction rate constant with singlet oxygen (1.08 × 10^8^ mol^−1^.L.s^−1^ < k < 1.2 × 10^8^ mol^−1^.L.s^−1^) [32,33]. The variation of FFA concentration in the natural waters under solar irradiation is illustrated in Figure 3 and the corresponding experimental degradation rate is given in Table 1.

The calculated steady state singlet oxygen concentrations are in accordance with the measured values on the surface of various freshwater and coastal waters (10^−11^ and 10^−14^ mol.L^−1^) [34]. [^1^O_2_]ss decreased from river ((13 ± 1) × 10^−14^ mol.L^−1^) to estuary ((8.8 ± 0.9) × 10^−14^ mol.L^−1^) and ocean water ((7.0 ± 0.7) × 10^−14^ mol.L^−1^), and may be related to the concentration variation in CDOM. The plot of the natural matrices’ absorbance at 250 nm, as a function of the steady state singlet oxygen concentration, showed a linear dependence (Figure 4), highlighting the correlation between singlet oxygen generation and CDOM.

Regarding riboflavin and porphine, in the presence of light and oxygen, they generate ROS that can be further involved in the oxidation of biological molecules such as mycosporines. Between pH 5.0 and 8, riboflavin (pK_1_ = 0.12 and pK_2_ = 9.95) [35] is under its neutral form with a singlet oxygen quantum yield of 0.6 at pH = 6.8 [36]. Porphine has two very close pKa values near pH 5.0, and for pH > 5.2 at low concentration, porphine exists in the monomeric deprotonated form [37] with a singlet oxygen quantum yield equal to 0.62 ± 0.3 [38]. The concentration of steady state singlet oxygen generated in pure water by porphine was estimated to be (7.8 ± 0.8) × 10^−11^ mol.L^−1^ for riboflavin and 3.2 times lower for porphine (2.4 ± 0.2) × 10^−11^ mol.L^−1^ (Table 1). The 1000 times higher values obtained for [^1^O_2_]ss in pure water compared to natural waters are related to a greater absorbance of photosensitizers compared to that of CDOM, especially in the visible domain. The singlet oxygen quantum yield (Φ_Δ_) (Equation (1)) represents the moles (N) of ^1^O_2_ produced per moles of photons absorbed by a sensitizer:(1)ΦΔ=N O21 formedN photons absorbed

The number of photons in mol absorbed by the sensitizers, calculated using their absorbance spectra and the lamp irradiance (Figure 5), was equal to 1.9 × 10^14^ for porphine, around five times lower than for riboflavin 9.1 × 10^14^, which is in the same order of magnitude of [^1^O_2_]ss.

### 2.2. Stability in Different Water Matrices

#### 2.2.1. Experiments Using Dark Controls

Before studying the M-ser(OH) and gadusolate photodegradation under solar irradiation conditions, dark controls were performed for river and estuary water by UV spectrophotometry.

Variation of the absorbance spectrum in the dark was observed for both gadusolate and M-ser(OH) (Figure 6). Such variation in the dark has already been reported for MAA dissolved in seawater [39]. However, degradation in the dark is low relative to the irradiated molecules, indicating that biotic degradation is relatively slow. Gadusolate is a less stable molecule than M-ser(OH) and more sensitive to biotic decomposition.

#### 2.2.2. Solar Irradiation

The pH, from 5.3 in pure water to 8.5 in river water, remained constant for the various aqueous matrices during the irradiation experiments. Gadusol with a pK_a_ at 4.25 was therefore in its negatively charged enolate form with a measured maximum absorption at 296 nm, as described by Arbeloa et al. [21]. An absorption maximum of 310 nm, in agreement with the values in the literature [14,19] was recorded for the neutral M-ser(OH) (Figure 5). The photodegradation of M-ser(OH) and gadusolate was followed by the spectral change, and Figure 7 illustrates the absorbance variation of these molecules under solar irradiation for six hours in pure and ocean water. No new bands appeared in the 200–800 nm range of the absorption spectra of the irradiated samples.

The photodegradation rates for the different water matrices were obtained from the slope of the linear fit of the absorbance data (at 296 nm for gadusolate and 310 nm for M-ser(OH)) as a function of irradiation time (Figure 8).

These rates were used to calculate the quantum yields (Figure 9, Table 2) from Equation (2), using the incident radiation intensity measured for the solar lamp (Figure 5).

The results show a similar behavior for both M-ser(OH) and gadusolate in all aqueous solutions with a high photostability (Φ_d_ < 20 × 10^−4^) which is, however, influenced by the matrices. Indeed, the quantum photodegradation yield decreases by more than one order of magnitude from river water Φ_d,Gd_ = (20 ± 6) × 10^−4^ and Φ_d,M-ser(OH)_ = (9 ± 1) × 10^−4^ to pure water Φ_d,Gd_ = (0.8 ± 0.2) × 10^−4^ and Φ_d,M-ser(OH)_ = (1.1 ± 0.6) × 10^−4^. The highest values are observed for natural waters and are proportional to the [^1^O_2_]ss measured for these matrices (Figure 10). Extrapolation from the regression line in the case of M-ser(OH) for [^1^O_2_]ss = 0 gives a value of (0.2 ± 00.4) which is close to that in pure and mineral water. The quantum yield for gadusolate is probably overestimated since a slight decomposition was observed in the dark but the slope difference observed seems to indicate a different reactivity with singlet oxygen between M-ser(OH) and gadusolate.

The quantum photodegradation yield of gadusolate and M-ser(OH) are (0.8 ± 0.2) × 10^−4^ and (1.1 ± 0.6) × 10^−4^ in pure water, and (0.6 ± 0.4) × 10^−4^ and (0.9 ± 0.2) × 10^−4^ in mineral water, respectively. These values are identical within experimental errors, showing a negligible role of ions in the photodegradation mechanism. It is slightly higher in artificial seawater (2.7 ± 0.6) × 10^−4^ for gadusolate and M-ser(OH) (1.6 ± 0.2) × 10^−4^ probably due to the pH increase from 7 to 8.2. Notably, De la Coba et al. [20] noticed a similar behavior with a decrease in the absorbance of M-ser(OH) dissolved in water of about 17% at pH 4 and 25% at pH 8.5 after 4.5 h of irradiation. Finally, it should be noted that the measured value for gadusolate in pure water (0.8 ± 0.2) × 10^−4^ is in agreement with that estimated by Arbeloa et al. [21] in air-saturated solution (1.27 ± 0.32) × 10^−4^.

#### 2.2.3. Selective Reaction with Photosensitizers

Photostability studies of these molecules under solar irradiation were completed in pure water using two photosensitizers (riboflavin RF and porphine PPY) able to produce ROS. These photosensitizers may participate in photooxidation depending on their triplet state formation via electron transfer (Type 1) or singlet oxygen production (Type 2) (Figure 11).

Thus, additional assays were performed in the presence of sodium azide, a quencher of singlet oxygen (^1^O_2_) [40], in order to identify the type of mechanism involved in the photodegradation by oxygen species (Figure 12). No degradation of the photosensitizers was observed in the presence of M-ser(OH) or gadusolate under our experimental conditions.

The photodegradation quantum yield (Figure 13, Table 2) increased in the presence of the photosensitizer, which is therefore involved in decomposition reactions. The presence of sodium azide that quenches singlet oxygen in water reduces this reaction by a factor of 10 for both sensitizer and M-ser(OH) or gadusolate. This confirms the previous results linking Φ_d_ to the amount of singlet oxygen present in natural matrices. However, this quantum yield is around five hundred times higher than in pure water, suggesting that the reactivity of gadusolate and M-ser(OH) with the sensitizer triplet is in line with the Type 1 mechanism (Figure 11).

Photolysis studies of mycosporines in the presence of flavins carried out by Bernillon et al. [23] showed different mechanisms of photodegradation according to mycosporine substituents. The preferred break site could not be identified in the case of M-ser(OH), but the side group of the cyclohexenone residue was suggested [23].

Orallo et al. [22] have determined the ^1^O_2_ reaction rate constant (k_Q_) of gadusolate to be 0.18 ± 0.08 × 10^8^ M^−1^s^−1^. Gadusolate reacts with triplet sensitizer ^3^Sens*, as a reductive quencher yielding the neutral radical gadusolate^●^ and the sensitizer anion radical Sens^●−^. This species can then transfer its newly acquired electron to oxygen present in the water, resulting in superoxide radical (O_2_^●−^) formation. Gadusolate and M-ser(OH) ion radicals can react with superoxide radicals to generate oxygenated peroxy radicals like GdOO^●^, MSerHOO^●^ [41,42]. The redox potential of gadusolate, molecular oxygen (^3^O_2_), triplet state of riboflavin and porphine was found to be 0.60 V [43], −0.33 V, 1.7 V [44] and 1.44 V [45], respectively. These reduction potential values make the Type 1 photosensitization thermodynamically favorable.

## 3. Materials and Methods

### 3.1. Biomolecules, Chemicals and Reagents

#### 3.1.1. Mycosporine-Serinol and Gadusol

Gadusol (Gd, C_8_H_12_N_2_O_6_, Ɛ_269nm_ (molar absorptivity) = 12,400 [15] or Ɛ_264nm_ = 12,900 [32] L.mol^−1^.cm^−1^, gadusolate Ɛ_296nm_ = 22,200 or 21,800 [15] or 22,750 [19] L.mol^−1^.cm^−1^, Figure 1) produced by yeasts was kindly offered by Prof. Taifo Mahmud from Oregon State University, USA. The sample was received in 2019 as a lyophilized pure extract and stored at −20 °C until use. Mycosporine-serinol (M-ser(OH), C_11_H_19_NO_6_, Ɛ_309nm_ = 25,516 [16] or Ɛ_310nm_ = 27,270 [15] L.mol^−1^.cm^−1^, Figure 1) extracted from the marine lichen *Lichina pygmaea* in the Andalusian coast in 2020 was purchased from the Laboratory of Photobiology of the Central Research Services of the University of Málaga (Malaga, Spain). M-ser(OH) was received as lyophilized pure extract and stored at −20 °C until use. The extraction, purification and characterization of the M-ser(OH) were performed as described previously [20].

#### 3.1.2. Matrices

Ocean water was collected from the Bay of Biscay (Anglet, France) in March 2021. Estuary water was collected in the Adour estuary (Bayonne, France) in March 2021. River water was collected from Gave de Pau (Pau, France) in April 2021. The natural water matrices were filtered using cellulose membrane filters with 0.45 µm pore size (from Merck) before the experiments. Pure water was obtained from a Millipore Milli-Q apparatus. Artificial seawater was prepared in house according to the standard norm ASTM D1141-98 (2013) [46] with the following main composition in percentages of each component measured by weight: NaCl (58.490%), MgCl_2_ (26.460%), Na_2_SO_4_ (9.750%), CaCl_2_ (2.765%) and KCl (1.645%), and pH adjusted to 8.2 using 0.1 mol.L^−1^ solution of sodium hydroxide or hydrochloric acid. Commercial mineral water was prepared with the following composition: [Ca]_tot_ = 80 mg.L^−1^, [Mg]_tot_ = 26 mg.L^−1^, [K]_tot_ = 1 mg.L^−1^, [Na]_tot_ = 6 mg.L^−1^, [NO_3_^2−^]_tot_ = 3.8 mg.L^−1^, [HCO_3_^−^]_tot_ = 360 mg.L^−1^, [SO_4_^2−^]_tot_ = 15 mg.L^−1^, [Si]_tot_ = 14 mg.L^−1^, [Cl^−^]_tot_ = 10 mg.L^−1^; pH = 7.2 was used.

#### 3.1.3. Photosensitizers

Riboflavin (RF, 7,8-Dimethyl-10-((2R,3R,4S)-2,3,4,5-tetrahydroxypentyl)benzo-[g]-pteridine-2,4-(3H,10H)-dione, 98% purity), porphine (PPY, (4,4′,4″,4‴-(porphine-5,10,15,20 tetrayl) tetrakis (benzenesulfonic acid) tetrasodium salt hydrate, 98% purity) and sodium azide (NaN_3_, >99.5%) were purchased from Merck, Germany. Both riboflavin and porphine are water soluble and have a series of absorption bands in the UV region and the visible region: riboflavin λ_Riboflavin(Max)_ = 222, 266, 373 and 447 nm, and porphine λ_Porphine (Max)_ = 413, 515, 552, 578 and 632 nm [47]. The most intense band is 413 nm and it is referred to as the Soret band, and the other four low-intensity bands correspond to the Q band. Both of these sensitizers possess a high molar absorptivity (Ɛ_RF_ = 13,222 L.mol^−1^.cm^−1^ at 447 nm pH = 7 [48], Ɛ_PPY_ = 21,900 L.mol^−1^.cm^−1^ at 414 nm). Furfuryl alcohol (C_5_H_6_O_2_; Merk) was freshly distilled before use and stored in the dark.

### 3.2. Determination of Singlet Oxygen Steady State Concentration in Natural Matrices

The steady state concentration of singlet oxygen ([^1^O_2_]_ss_) was determined by the indirect method of Haag and Hoigne [33,49] using quenching experiments. Water matrices and photosensitizers capable of generating singlet oxygen (^1^O_2_) or radical species (O_2_.^−^) under UV light ([PPY] = 4 × 10^−7^ mol.L^−1^; [RF] = 1 × 10^−6^ mol.L^−1^) [50,51] were irradiated in the presence of furfuryl alcohol ([FFA] = 3.1 × 10^−5^ mol.L^−1^). The major product of singlet oxygen reaction with FFA is 6-hydroxy-(2H)-pyran-3-one (6-HP-one) and the experimental rate constant was deduced from the slope of a linear plot of ln([FFA]_0_/[FFA]_t_) versus time for the first-order decay process. [^1^O_2_]ss was calculated by dividing the experimental degradation rate constant of furfuryl alcohol as a function of irradiation time by the recommended value for k_(O2+FFA)_ = 1.08 × 10^8^ mol^−1^.L.s^−1^, which is corrected for temperature and salt content [52]. FFA concentration was quantified by high-performance liquid chromatography (HPLC Agilent 1290 Infinity II, Santa Clara, CA, USA). HPLC analysis was carried out with a Supelco Lichrosphere RP18-5 (25 mm × 4.6 mm, 5 µm) column, 20 µL injection, eluent 80% water with 0.1% H_3_PO_4_, 20% acetonitrile, rate 2 mL.min^−1^, UV detection at 205 nm and 218 nm. The retention times of FFA and 6-HP-one were 2.9 min and 1.8 min, respectively. No new chromatographic peaks for hypothetical adducts (i.e., of either pyruvic acid (PA) or its photolytic radicals with FFA) could be observed, discarding their direct reactivity with FFA. The natural water matrices (ocean, estuary, river water) were irradiated for time periods of 0.5 and 1 h for river water, 2 and 4 h for ocean water, and 1, 2 and 4 h for estuary water. The photosensitizers in pure water were irradiated for 1 min for riboflavin and 3 min for porphine. All the experiments were conducted in triplicate.

### 3.3. Decomposition Experiments

#### 3.3.1. Solar Irradiation

The different solutions were irradiated in a Sun Test XLS solar simulator from Atlas Material Testing Solutions (Mount Prospect, IL, USA) with a horizontal 1170 cm^2^ exposure area, an air-cooled 1700 W xenon lamp and a daylight filter. The irradiations were performed with an irradiance of 590 W.m^−2^ and a dosage per hour of 2000 KJ.m^−2^ in the wavelength range of 250 to 700 nm at 30 °C. The above experimental conditions were chosen to fit the conditions of natural solar radiation [53]. The total time of irradiation was chosen as 6 h (total dosage of 12,000 KJ.m^−2^). The absorption spectra of the different solutions were recorded using a Perkin Elmer UV/VIS/NIR Lambda-750 Spectrophotometer (Waltham, MA, USA) at subsequent time intervals every 2 h during the irradiation experiment to monitor the kinetics of photodegradation. These experiments were conducted in triplicate. An Avaspec 2048L spectroradiometer (Avantes, Apeldoorn, The Netherlands) with a 1 m × 600 µm UV optic fiber and a 3900 µm 180° cosinus sensor was used for irradiance measurements. M-ser(OH) or gadusolate was dissolved in various water matrices (pure, mineral, artificial seawater, river, estuary and ocean) with an absorbance fixed around 1 at the maximum wavelength of absorption for M-ser(OH) or gadusolate or gadusol and for an optical path of 0.2 cm. A 0.6 mL volume of solution, inserted into the Hellma cuvet (height: 3 cm, depth: 1 cm and optical path: 0.2 cm) closed by a teflon cap, was used for the irradiation experiments. The photostability of M-ser(OH) and gadusolate was studied in the presence of riboflavin and porphine under the same irradiation conditions. In the presence of riboflavin, the gadusolate and M-ser(OH) solutions were irradiated for 20 and 60 s, and their corresponding absorption spectra were analyzed in a range of 5 and 10 s, respectively. Similarly, M-ser(OH) and gadusolate solutions were both irradiated in the presence of porphine for a total time period of 5 min. The spectra were registered at every time interval during the irradiation experiments. Additionally, to evaluate the respective fraction of M-ser(OH) or gadusolate degradation by reactive oxygen species in pure water, sodium azide (5 × 10^−3^ mol.L^−1^) was used as a physical quencher of singlet oxygen formed by riboflavin or porphine under irradiation. Thus, the photodegradation of M-ser(OH) or gadusolate was conducted for a total irradiation time of 20 s. Their corresponding spectra were analyzed every 5 s. Similarly, M-ser(OH) and gadusolate were both irradiated in the presence of porphine for a total time period of 5 min. The spectra were registered at every time interval during the irradiation experiments. The concentrations of M-ser(OH), gadusolate and photosensitizers are given in SI.

#### 3.3.2. Dark Controls

Dark controls were realized for river and estuary water to evaluate the biotic degradation when keeping the sample for 6 h without solar irradiation. These experiments were performed in triplicate. The absorption was measured after the 6 h.

### 3.4. Photodegradation Quantum Yield Determination

The photostability of the compounds was compared using quantum yield of photodegradation (Φ_d_) defined as in Equation (2):(2)$$\Phi_{d}=\frac{N_{phM}(number\;of\;photodegraded\;molecules)}{N_{v_{Abs}}(number\;of\;absorbed\;photons)}$$

*N*_phM_ = r_o_ × V × N_A×_t, where r_o_ is the initial photodegradation rate (mol.L^−1^.s^−1^), V is the volume of the irradiated solution (L), N_A_ is the Avogadro number (6.022 × 10^23^ mol^−1^) and t is the irradiation time (s). Where (Equation (3)):(3)$$N_{v_{Abs}}=S\times t\times \sum_{\lambda min}^{\lambda max}P_{0,\lambda i}\times (1-10^{A_{\lambda i}})$$
where *S* is the irradiated surface of the solution (cm^2^), t is the irradiation time (s), *P*_0,λi_ the photon flux absorbed by the M-ser(OH) or gadusolate solutions at the wavelength *λ_i_* (number of photon emitted.s^−1^.cm^−2^), *λ_min_* and *λ_max_* are the wavelength range of the measurement, and *A_λi_* the absorbance of the M-ser(OH) or gadusolate solutions at the wavelength *λ_i_*.
(4)$$\Phi_{d}=\frac{r_{0}\times V\times N_{A}}{S\times \sum_{\lambda min}^{\lambda max}P_{0,\lambda i}\times (1-10^{A_{\lambda i}})}$$

The initial photodegradation rate, *r*_0_, was determined from the slope of the linear regression of the plots of the concentration (calculated from the maximum absorbance and their corresponding molar absorption coefficient Ɛ_309nm_ = 25,516 [14] L.mol^−1^.cm^−1^ for M-ser(OH) and Ɛ_296nm_ = 21,800 [13] L.mol^−1^.cm^−1^ for gadusolate and path length = 0.2 cm) versus irradiation time. *P*_0,*λi*_ was measured with an Avaspec 2048L spectroradiometer (Avantes B.V.; Apeldoorn; The Netherlands).

### 3.5. Statistical Analysis

The results were expressed as mean ± standard deviation (SD). All experiments were conducted in triplicate.

## 4. Conclusions

In comparison to existing conventional sunscreens, M-ser(OH) and gadusolate are of natural origin and are known to present high photostability. In the present study, the photodegradation of both M-ser(OH) and gadusolate was assessed in different water matrices and in the presence of photosensitizers. Here, it was demonstrated that the photodegradation of both M-ser(OH) and gadusolate was higher in natural matrices (river, estuary and ocean) than in pure water due to the generation of ROS through irradiation of dissolved colored organic matter. Also, the photodegradation was highest in river water and lowest in ocean water. Interestingly, it was also shown that M-ser(OH) was more photostable than gadusolate, probably due to the presence of the side moiety. The present work emphasizes the ability of mycosporine-serinol and gadusolate to withstand photodegradation, and supports their role as effective and stable UV-B filters.

## Figures and Tables

**Figure 1 marinedrugs-22-00473-f001:**
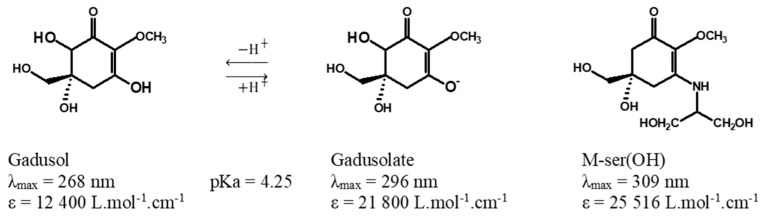
Chemical structures of gadusol, gadusolate [17,18] and mycosporine-serinol (M-ser(OH)) [19].

**Figure 2 marinedrugs-22-00473-f002:**
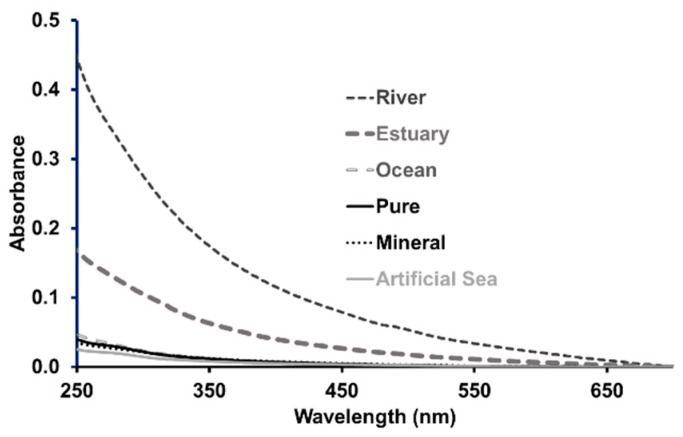
Absorbance profiles of the aqueous matrices as a function of wavelength at room temperature (l = 1 cm).

**Figure 3 marinedrugs-22-00473-f003:**
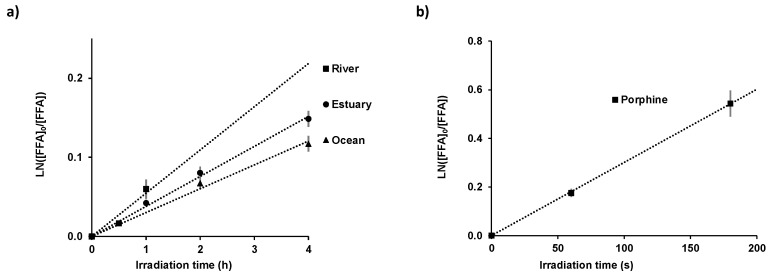
Variation of Ln([FFA]_0_/[FFA]) as a function of solar irradiation time for the natural matrices, namely, river, estuary and ocean (**a**); and for porphine in pure water at room temperature, [FFA]_0_ = 3.1 × 10^−5^ mol.L^−1^ (**b**). The values are presented as mean ± standard deviation, *n* = 3.

**Figure 4 marinedrugs-22-00473-f004:**
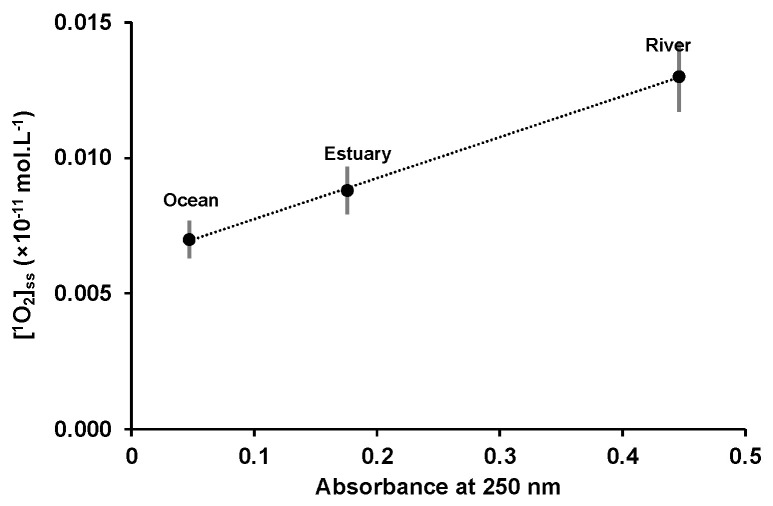
Variation of [^1^O_2_]ss as a function of the absorbance at 250 nm of the natural waters at room temperature, l = 1 cm. The values are presented as mean ± standard deviation (*n* = 3).

**Figure 5 marinedrugs-22-00473-f005:**
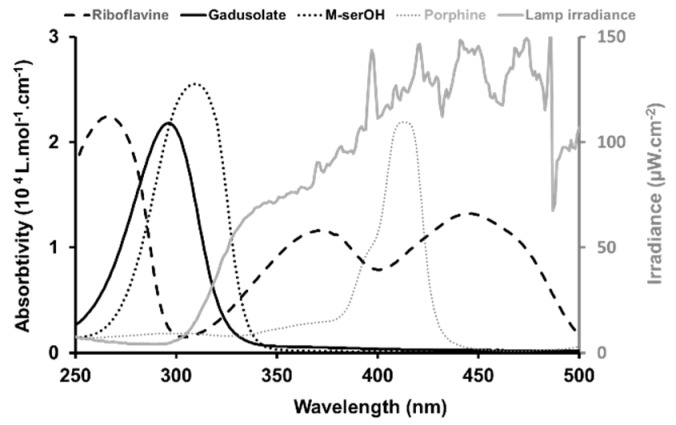
Overlay of the electronic spectra of gadusolate, M-ser(OH) riboflavin and porphine (left scale) and spectral irradiance of the lamp used in the solar simulator (right scale).

**Figure 6 marinedrugs-22-00473-f006:**
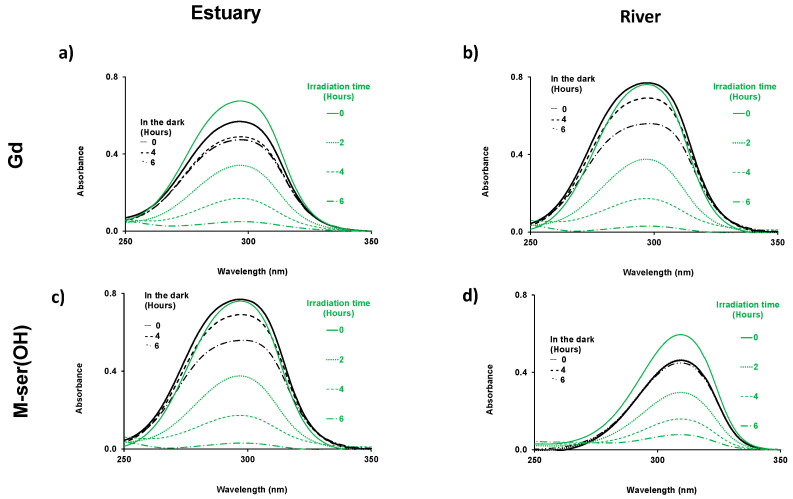
Time-dependent absorption spectra of gadusolate ([Gd]_0_ = 1.5 × 10^−4^ mol.L^−1^ in estuary and 1.7 × 10^−4^ mol.L^−1^ in river water) (**a**,**b**); and M-ser(OH) ([M-ser(OH)]_0_ = 1.3 × 10^−4^ mol.L^−1^ in estuary and 1.2 × 10^−4^ mol.L^−1^ in river water) (**c**,**d**), in the dark and under irradiation in estuary and in river water for 6 h, l = 0.2 cm.

**Figure 7 marinedrugs-22-00473-f007:**
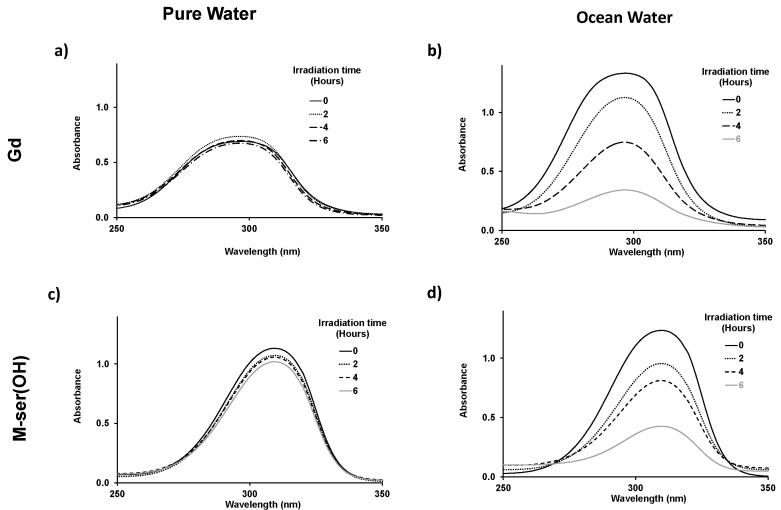
Time-dependent absorption spectra of gadusolate ([Gd]_0_ = 1.7 × 10^−4^ mol.L^−1^ in pure water and 3.1 × 10^−4^ mol.L^−1^ in ocean) (**a**,**b**) and M-ser(OH) ([M-ser(OH)]_0_ = 2.2 × 10^−4^ mol.L^−1^ in pure water and 2.4 × 10^−4^ mol.L^−1^ in ocean) (**c**,**d**), in pure water and ocean after 0, 2, 4, 6 h of irradiation; l = 0.2 cm.

**Figure 8 marinedrugs-22-00473-f008:**
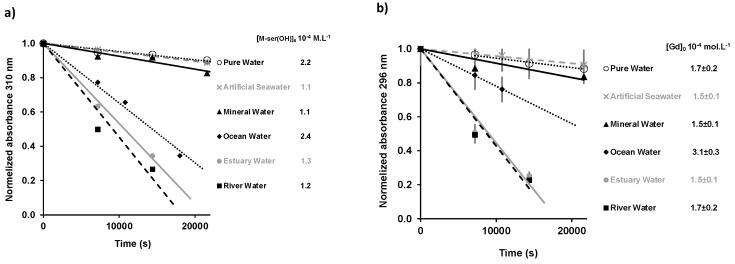
Variation of maximum absorbance relative to the initial absorbance as a function of the irradiation time in different aqueous matrices (M-ser(OH) at 310 nm (**a**) gadusolate at 296 nm (**b**)). The values are presented as mean ± standard deviation, *n* = 3.

**Figure 9 marinedrugs-22-00473-f009:**
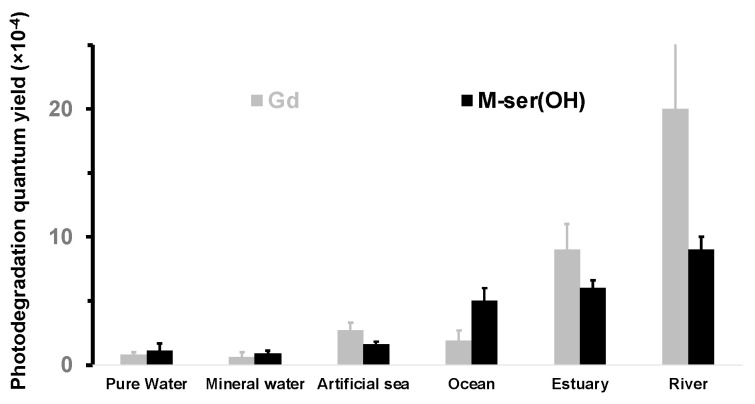
Photodegradation quantum yields of gadusolate and M-ser(OH) in aqueous matrices under solar irradiation (values are presented as mean ± standard deviation, *n* = 3).

**Figure 10 marinedrugs-22-00473-f010:**
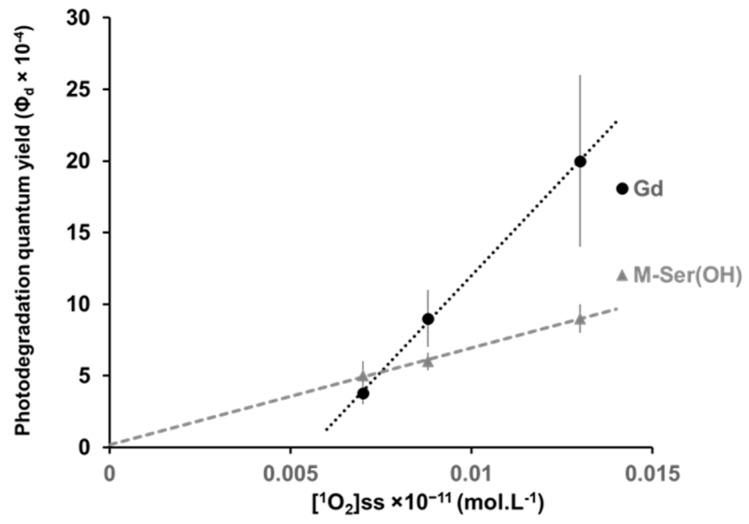
Variation of the photodegradation quantum yield under solar irradiation relative to the initial absorbance as a function of the [^1^O_2_]_ss_ measured in natural matrices (ocean, estuary, river). The values are presented as mean ± standard deviation (*n* = 3).

**Figure 11 marinedrugs-22-00473-f011:**
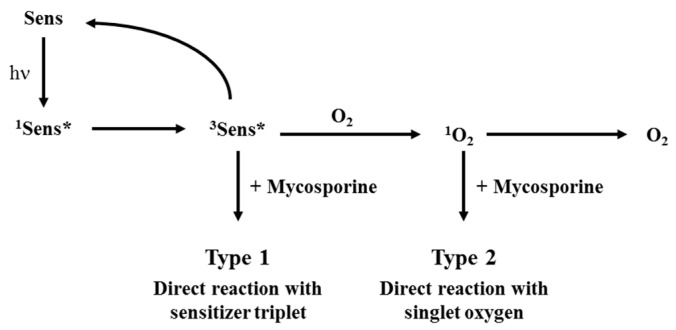
Sensitized photodegradation pathways (Sens (sensitizer ground state), ^1^Sens* (sensitizer first excited state) are ^3^Sens* sensitizer triplet state). Note: in the figure, ‘+Mycosporine’ is related to M-ser(OH) or gadusolate.

**Figure 12 marinedrugs-22-00473-f012:**
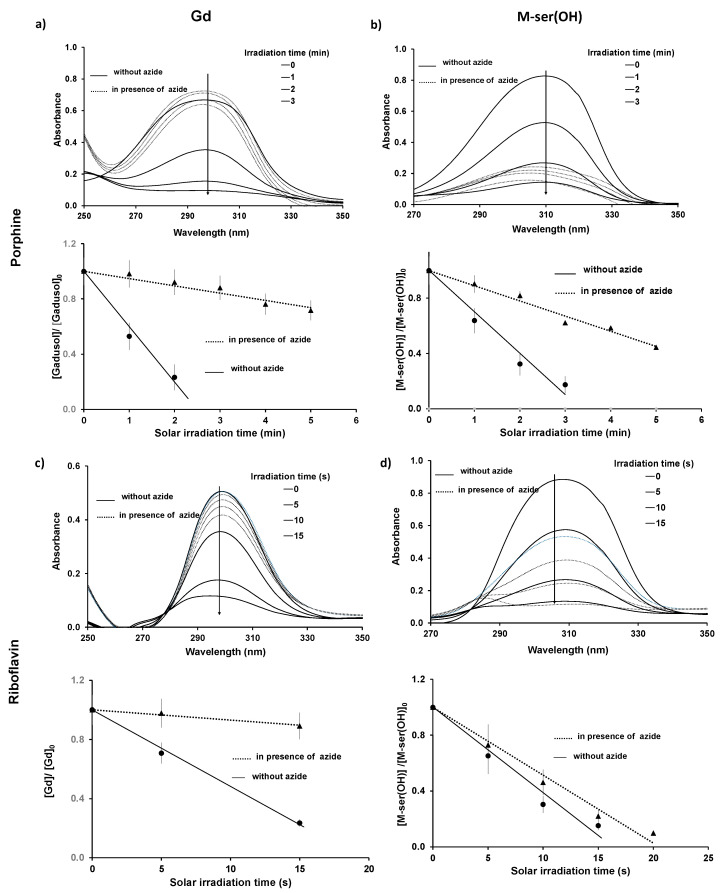
Time-dependent absorption spectra of gadusolate (1 × 10^−4^ mol.L^−1^ < [Gd]_0_ < 1.8 × 10^−4^ mol.L^−1^) and M-ser(OH) (4.7 × 10^−5^ mol.L^−1^ < [M-ser(OH)]_0_ < 1.7 × 10^−4^ mol.L^−1^) after 0, 1, 2 and 3 min of irradiation for porphine (2.0 × 10^−4^ mol.L^−1^) (**a**,**b**), and after 0, 5, 10, 15 s of irradiation for riboflavin (4.0 × 10^−5^ mol.L^−1^) (**c**,**d**), l = 0.2 cm. The values are presented as mean ± standard deviation, *n* = 3.

**Figure 13 marinedrugs-22-00473-f013:**
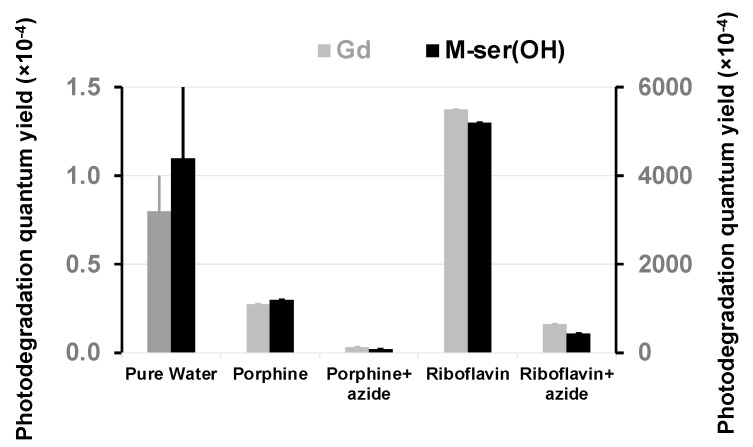
Photodegradation quantum yields of gadusolate and M-ser(OH) in water (left scale) and in the presence of photosensitizers and sodium azide (right scale) under solar irradiation (values are presented as mean ± standard deviation, *n* = 3).

**Table 1 marinedrugs-22-00473-t001:** Total irradiation time, rate phototransformation of FFA ([FFA]_0_ = 3.1 × 10^−5^ mol.L^−1^) and steady state singlet oxygen concentration in natural matrices and in pure water in the presence of photosensitizer ([Riboflavin] = 1.34 × 10^−5^ mol.L^−1^; [Porphine] = 4 × 10^−7^ mol.L^−1^).

Matrices	Total Irradiation Time (h)	k_app_ × 10^−3^ (s^−1^)	[^1^O_2_]_ss_ × 10^−11^ (mol.L^−1^)
River Water	1	0.015 ± 0.002	0.013 ± 0.001
Estuary Water	4	0.0105 ± 0.0002	0.0088 ± 0.0009
Ocean Water		0.00084 ± 0.0003	0.0070 ± 0.0007
Photosensitizer in pure water	Total Irradiation Time (s)	k_app_ × 10^−3^ (s^−1^)	[^1^O_2_]_ss_ × 10^−11^ (mol.L^−1^)
Riboflavin	60	9.3 ± 0.9	7.8 ± 0.8
Porphine	180	2.89 ± 0.02	2.4 ± 0.2

Rate phototransformation of FFA and steady state singlet oxygen concentrations are presented as mean ± standard deviation, *n* = 3.

**Table 2 marinedrugs-22-00473-t002:** Photodegradation quantum yield of gadusolate and M-ser(OH) in the presence of porphine and riboflavin and sodium azide.

Natural Compounds	Photodegradation Quantum Yield (Φ × 10^−4^)								
	Pure Water				Mineral Water	Artificial Seawater	Ocean Water	Estuary Water	River Water
Gadusolate	0.8 ± 0.2				0.6 ± 0.4	2.7 ± 0.6	2.9 ± 0.8	9 ± 2	20 ± 6
	Porphine		Riboflavin						
	without azide	with azide	without azide	with azide					
	1100 ± 200	130 ± 30	5500 ± 1000	650 ± 100					
M-Serinol	1.1 ± 0.6				0.9 ± 0.2	1.6 ± 0.2	5 ± 1	6.0 ± 0.6	9 ± 1
	Porphine		Riboflavin						
	without azide	with azide	without azide	with azide					
	1200 ± 200	140 ± 30	5200 ± 1000	440 ± 90					

Photodegradation quantum yields are presented as mean ± standard deviation, *n* = 3.

## Data Availability

The data presented in this study are available on request from the corresponding author.

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
