# Peer review of "Effect of Reactive Oxygen Species Photoproduced in Different Water Matrices on the Photostability of Gadusolate and Mycosporine-Serinol"

_marinedrugs, 2024, doi:10.3390/md22100473_

Round 1
Reviewer 1 Report
Comments and Suggestions for Authors
This work describes an experimental study on the photostability of two natural UV-absorbing molecules: mycosporine serinol (Myc) and gadusolate (Gd, the enolate form of gadusol), in different aqueous matrices, including samples of natural water.
First of all, it must be said that the title itself is confusing since it refers to gadusol as belonging to the category of mycosporine compounds and this is wrong. Gadusols have been previously described as structural- and biosynthetically related to mycosporines. But, whereas mycosporines include an amino acid moiety in their chemical structure, gadusols don´t. Similarly, in the introduction section, the criterium used for the distinction between mycosporine as UV-B absorbing metabolites, and mycosporine-like amino acids (MAAs) as UV-A absorbing compounds, is not correlated with the definitions found in the classical references of the subject within the fields of photobiology or natural products. Thus, I recommend the authors to revise these notions in founding papers by Bandaranayake (Nat. Prod. Rep., 1998, 15, 159-172) and by Shick & Dunlap (Annu Rev Physiol. 2002:64:223-62).
The goals of this study seem to be oriented to provide the basic knowledge needed to understand the role and evaluate the application of the natural molecules in sunscreen formulation as alternatives to synthetic UV-filters. The photostability of various MAAs and gadusol has already been assessed. However, there are no previous studies on the photoinduced decomposition of mycosporine-serinol and the enolate form of gadusol in natural waters.
However, the meaning of the author´s findings concerning the purpose of evaluating the performance of the natural metabolites in photoprotection is hardly outlined in the manuscript, contrasting the affirmations in lines 18-20 and 387-388.
Even more worrying is that the experimental design and the reported data show serious deficiencies that compromise the significance of this research, as it is explained below.
Photodegradation experiments were carried out with a solar simulator irradiation source which emits from 250 nm to 700 nm, however, the distribution of sun irradiation arriving at the Earth's surface comprises wavelengths larger than 290 nm. Thus, the interval between 250 and 290 nm does not represent the real environmental condition. This is important because the absorption spectra of the metabolites here studied, as well as those of the photosensitizers, may partially overlap this portion of the irradiation range.
The calculation of the photodecomposition quantum yield of a species X, as eq. 1 correctly indicates, involves the evaluation of the number of photons absorbed by X, i.e. the absorbance of X. Thus, it is essential to describe how these absorbances were determined, particularly when mixtures of substances are irradiated, such as those samples prepared with river, estuary or sea water? Moreover, according to Fig. 2, the absorbance of these “solvents” is far from negligible in the spectral region of the irradiation. In this sense, it is relevant to contemplate the treatment and quality of the samples of natural water used here. Were they previously filtered? Are they homogeneous or colloidal systems instead? It is important to notice that the spectra shown in Fig.2 may also account for light scattering, not only for the absorbance of CDOM. Both should be adequately distinguished from the absorbance of the X species (Myc or Gd) to evaluate quantum yields correctly through eq 3. This also raises questions about the meaning of the results in Fig. 4 where the values of the “absorbance at 250 nm” of the natural waters are said to be considered.
Likewise, in the experiments with photosensitizers, it is not clarified how the absorption of the irradiation is distributed and, unfortunately, Figure 5 is missed. Which species is/are mainly absorbing the photons, Myc or Gd vs. photosensitizers? Were the experiments with riboflavin and with porphine carried out in equivalent conditions of light absorptions?
It is disconcerting the use of dissimilar conditions, such as the initial concentrations of Gd and Myc in some series of experiments, for example in Fig.7, thus avoiding an appropriate comparison of the reactivities.
The description of the experiments with the presence of sodium azide also omits to inform the initial concentration of this compound in the irradiated samples. Azides generally absorb in the UV-B region (see Mc Donald et al. Journal of Chemical Physics 52, 1332 (1970))
All these issues should be carefully analyzed and revised to take into account or neglect possible inner filter effects before properly comparing the results and meeting the requirement of a consistent experimental design.
In the opinion of this reviewer, there are further weaknesses that should be considered in reformulating this work.
-Riboflavin as a photosensitizer not only generates singlet oxygen but also superoxide anion radical however the possible effect on the photodecomposition of Myc or Gd are exclusively discussed in terms of singlet oxygen (see for instance Fig.11)
-The plot for riboflavin in Figure 3 includes only 2 experimental points and still a linear correlation is proposed, which is not reliable.
-It is concluded on page 12 that the results in Figure 13 and Table 2, prove that Type I mechanisms explain the reactivity of Gd and Myc with the sensitizer triplet. This is quite speculative and more experiments in the absence of air and/or using direct identification of the intermediates (for example, laser flash photolysis, or EPR) should be conducted to support this affirmation.
-Analogously, the discussion outlined in lines 373-376 lacks references that may provide stronger evidence, and the conclusion about the comparative photostability of Myc against Gd based on “its bulkier and robust structure” has no clear support.
-The sources of some kinetic and spectrophotometric parameters cited in the manuscript seem to be wrongly assigned. Namely:
- the 1O2 quenching rate constant (kQ) by Gd in water is not reported by Losantos et al (ref.# 49) and it does not amount 2x108 M-1s-1 (see lines 370 and on)
-Arbeloa´s paper does not correspond to ref.#16
-The redox potential for Gd has not been reported by Shick et al. 2002 (ref # 50). Actually, I feel curious about where the value of 0.60 V comes from.
-Verify correspondence of ref# 14 in line 89; ref# 15 in lines 94 and 332
Other minor points:
-PA is not defined (line 135)
-“Porphyrine” in Table 2
- Eq. 2: lambda 1 and lambda 2 are not defined
-In lines 258 and 260: singlet quantum yields?, or singlet oxygen quantum yields?
-Fig. 8: revise units (M.L-1?)
-line 366: d-carbonyl group?
Comments on the Quality of English Language
The entire manuscript must be reviewed for consistency of verbs, vocabulary and spelling errors.
Author Response
|
Comments 1: This work describes an experimental study on the photostability of two natural UV-absorbing molecules: mycosporine serinol (Myc) and gadusolate (Gd, the enolate form of gadusol), in different aqueous matrices, including samples of natural water. First of all, it must be said that the title itself is confusing since it refers to gadusol as belonging to the category of mycosporine compounds and this is wrong. Gadusols have been previously described as structural- and biosynthetically related to mycosporines. But, whereas mycosporines include an amino acid moiety in their chemical structure, gadusols don´t. |
|
Response 1: The title of the paper has been modified as desired by the referee. We accept the fact that gadusols are not mycosporine-like compounds. We have made the required changes through the manuscript. This mistake occurred due to the fact that Gadusols have been previously described as structural- and biosynthetically related to mycosporines. We appreciate the very critical comment of the referee. Page 1 Line 3 has been replaced by photostability of gadusol, Page 2 Line 55 has been replaced by Chemical structure of: gadusol, gadusolate, Page 2 Line 83 as Mycosporines and gadusol, Page 13 In table 2 mycosporines have been replaced by materials. |
|
Comments 2: Similarly, in the introduction section, the criterium used for the distinction between mycosporine as UV-B absorbing metabolites, and mycosporine-like amino acids (MAAs) as UV-A absorbing compounds, is not correlated with the definitions found in the classical references of the subject within the fields of photobiology or natural products. Thus, I recommend the authors to revise these notions in founding papers by Bandaranayake (Nat. Prod. Rep., 1998, 15, 159-172) and by Shick & Dunlap (Annu Rev Physiol. 2002:64:223-62). |
|
Response 2: As suggested by the referee, the introduction part has been carefully revised by correlating with the definitions found in the classical references of the subject within the fields of photobiology or natural products. The papers of Bandaranayake, and Shick & Dunlap have been cited in the revised manuscript. Page 1 Line 29-35 has been replaced by, Mycosporines are UV absorbing compounds at around 310 nm which possesses a cy-clohexenone ring system linked with an amino acid or an amino alcohol. On the other hand, MAA are UV absorbing compounds with imine derivatives of mycosporines or enamino imines that consists of a cyclohexenimine ring system with UV absorption maxima between 310 and 360 nm. Both possess high molar extinction coefficients (from 28100 to 60 000 L.mol-1.cm-1), ability to dissipate the absorbed UV-radiation as heat energy without generation of oxidative photo-products, and high photo-and thermostability. Page 1 Line 37-38 has been replaced by: Mycosporines and MAAs are low-molecular weight (< 400 Da) and water-soluble molecules. Comments 3: The goals of this study seem to be oriented to provide the basic knowledge needed to understand the role and evaluate the application of the natural molecules in sunscreen formulation as alternatives to synthetic UV-filters. The photostability of various MAAs and gadusol has already been assessed. However, there are no previous studies on the photoinduced decomposition of mycosporine-serinol and the enolate form of gadusol in natural waters. However, the meaning of the author´s findings concerning the purpose of evaluating the performance of the natural metabolites in photoprotection is hardly outlined in the manuscript, contrasting the affirmations in lines 18-20 and 387-388. Response 3: As suggested by the referee, the purpose of evaluating the performance of the natural metabolites in photoprotection has been outlined in the revised manuscript, in agreement with the affirmations in lines 18-20 and 387-388 of the revised manuscript. Page 1 Line 17-19: These could be potential candidates to be used as UV-B filters in sunscreen formulations due to its biobased and ecofriendly nature, inherent biodegradability in contrary to the current existing conventional sunscreens. Page 14 Line 398-401: In conclusion, due to its biobased, ecofriendly nature and inherent biodegradability in natural environments in contrary to the current existing conventional sunscreens these could be potential candidates to be used as UV-B filters in sunscreen formulations. Comments 4: Photodegradation experiments were carried out with a solar simulator irradiation source which emits from 250 nm to 700 nm, however, the distribution of sun irradiation arriving at the Earth's surface comprises wavelengths larger than 290 nm. Thus, the interval between 250 and 290 nm does not represent the real environmental condition. This is important because the absorption spectra of the metabolites here studied, as well as those of the photosensitizers, may partially overlap this portion of the irradiation range. Response 4: As suggested by the referee, we have considered this point; The wavelength ranging from 250 to 290 belongs to the UV-C region. The assumption is that almost 99.9 % is absorbed by the ozone layer and 0.1 % gets transmitted across the atmosphere due to ozone layer depletion. All the spectra have been normalized accordingly. Comments 5: The calculation of the photodecomposition quantum yield of a species X, as eq. 1 correctly indicates, involves the evaluation of the number of photons absorbed by X, i.e., the absorbance of X. Thus, it is essential to describe how these absorbances were determined, particularly when mixtures of substances are irradiated, such as those samples prepared with river, estuary or sea water? Moreover, according to Fig. 2, the absorbance of these “solvents” is far from negligible in the spectral region of the irradiation. In this sense, it is relevant to contemplate the treatment and quality of the samples of natural water used here. Were they previously filtered? Are they homogeneous or colloidal systems instead? It is important to notice that the spectra shown in Fig.2 may also account for light scattering, not only for the absorbance of CDOM. Both should be adequately distinguished from the absorbance of the X species (Myc or Gd) to evaluate quantum yields correctly through eq 3. This also raises questions about the meaning of the results in Fig. 4 where the values of the “absorbance at 250 nm” of the natural waters are said to be considered. Response 5: As suggested by the referee we have looked into this point: The absorbance of these compounds (e.g., in river water) were determined by taking river water as reference (normalized). These waters were filtered to make the scattering parameter negligible. In Figure 2, the absorbance of these aquatic matrices was determined by using pure millique water as reference. These were found to be homogeneous solutions. As these solutions were filtered, we assume that it is due to CDOM. The absorbance at 250 nm for natural waters was not considered to evaluate the absorbance. Comments 6: Likewise, in the experiments with photosensitizers, it is not clarified how the absorption of the irradiation is distributed and, unfortunately, Figure 5 is missed. Which species is/are mainly absorbing the photons, Myc or Gd vs. photosensitizers? Were the experiments with riboflavin and with porphine carried out in equivalent conditions of light absorptions? Response 6: As suggested by the referee we have looked into it this point. We have included Figure 5 in the manuscript Photons are absorbed by gadusol, M-serinol and photosensitizers (riboflavin-266, 373, 447 nm and porphine-310, 414, 517 nm)]. The experiments with riboflavin and porphine were carried out in equivalent conditions of light absorptions. Both these photosensitizers were irradiated (250- 700 nm). The spectra were normalized accordingly. Comments 7: It is disconcerting the use of dissimilar conditions, such as the initial concentrations of Gd and Myc in some series of experiments, for example in Fig.7, thus avoiding an appropriate comparison of the reactivities. The description of the experiments with the presence of sodium azide also omits to inform the initial concentration of this compound in the irradiated samples. Azides generally absorb in the UV-B region (see Mc Donald et al. Journal of Chemical Physics 52, 1332 (1970)) Response 7: As suggested by the referee we have looked into this point. The initial concentration of Gd and Myc are different. But quantum yield of photodegradation is independent of the initial concentration. The UV absorption of azides was taken into account and subtracted from the spectra of the compounds post irradiation. Also, the initial concentration of sodium azide (5.10-3 mol.L-1) is given in the experimental part. Comments 8: Riboflavin as a photosensitizer not only generates singlet oxygen but also superoxide anion radical however the possible effect on the photodecomposition of Myc or Gd are exclusively discussed in terms of singlet oxygen (see for instance Fig.11) -The plot for riboflavin in Figure 3 includes only 2 experimental points and still a linear correlation is proposed, which is not reliable. Response 8: Thanks for your comments. The referee is right, the second part of the figure 3 has been removed. However, the values of kapp and [O2]ss in table 1 are still valid. Our results strongly suggest that the type 2 mechanism (singlet oxygen mechanism) is predominant in the degradation of the compounds studied. However, when a quencher of singlet oxygen is present in high concentration, degradation occurs at a lower rate, leading us to propose that the type 1 mechanism is also involved. Comments 9: It is concluded on page 12 that the results in Figure 13 and Table 2, prove that Type I mechanisms explain the reactivity of Gd and Myc with the sensitizer triplet. This is quite speculative and more experiments in the absence of air and/or using direct identification of the intermediates (for example, laser flash photolysis, or EPR) should be conducted to support this affirmation. Response 9: The referee is absolutely right. However, this study was mainly devoted to the behaviour of gadusolate and serinol in a natural environment and environmental conditions and the primary objective was not to carry out a photophysical study. Comments 10: Analogously, the discussion outlined in lines 373-376 lacks references that may provide stronger evidence, and the conclusion about the comparative photostability of Myc against Gd based on “its bulkier and robust structure” has no clear support. Response 10: Our results clearly indicate that there is a difference in stability between the two compounds studied. We have changed the wording in the conclusion to state that ‘This study infers that M-ser(OH) was more photostable relative to gadusolate probably due to the presence of the bulky (1,3-dihydroxypropan-2-yl) moiety’. Comments 11:-The sources of some kinetic and spectrophotometric parameters cited in the manuscript seem to be wrongly assigned. Namely: the 1O2 quenching rate constant (kQ) by Gd in water is not reported by Losantos et al (ref.# 49) and it does not amount 2x108 M-1s-1 (see lines 370 and on) -Arbeloa´s paper does not correspond to ref.#16 -The redox potential for Gd has not been reported by Shick et al. 2002 (ref # 50). Actually, I feel curious about where the value of 0.60 V comes from. -Verify correspondence of ref# 14 in line 89; ref# 15 in lines 94 and 332 Other minor points: -PA is not defined (line 135) -“Porphyrine” in Table 2 - Eq. 2: lambda 1 and lambda 2 are not defined -In lines 258 and 260: singlet quantum yields?, or singlet oxygen quantum yields? -Fig. 8: revise units (M.L-1?) -line 366: d-carbonyl group? Response 11: Thanks for your comments. The necessary changes have been made in the manuscript Page 14 Line 382-383, has been modified to Orallo et al. have determined that the 1O2reaction rate constant (kQ) of gadusolate to be 0.18 ± 0.08 × 108 M-1s-1 and reference 51 has been changed to Orallo, D. E., Lores, N. J., Arbeloa, E. M., Bertolotti, S. G., & Churio, M. S. (2020). Sensitized photo-oxidation of gadusol species mediated by singlet oxygen. Journal of Photochemistry and Photobiology B: Biology, 213, 112078 Page 14 Line 388, The reference 53 have been changed to: Baptista, Mauricio S., Jean Cadet, Alexander Greer, and Andres H. Thomas. "Photosensitization reactions of biomolecules: Definition, targets and mechanisms." Photochemistry and Photobiology 97, no. 6 (2021): 1456-1483. Aveline, Béatrice M. "Primary processes in photosensitization mechanisms." Comprehensive Series in Photosciences. Vol. 2. Elsevier, 2001. 17-37 Page 14 Line 389, the reference 54 has been changed to Arbeloa, E. M., Ramírez, C. L., Procaccini, R. A., & Churio, M. S. (2012). Electrochemical characterization of the marine antioxidant gadusol. Natural Product Communications, 7(9), 1934578X1200700928 Page 3 Line 100, the reference 14 has been changed to 26 Fernandes, S.C.M.; Alonso-Varona, A.; Palomares, T.; Zubillaga, V.; Labidi, J.; Bulone, V. Exploiting mycosporines as natural molecular sunscreens for the fabrication of UV-absorbing green materials. ACS Applied Materials & Interfaces 2015, 7, 30, 16558-16564.
Page 3 Line 96, reference 15 has been changed to 28 Bens, G. Sunscreens. In: Sunlight, Vitamin D and Skin Cancer. Springer, New York, NY. 2014, 429-463 Page 11 Line 340, reference 15 has been changed to 52 De la Coba, F.; Aguilera, J.; Korbee, N.; de Gálvez, M.V.; Herrera-Ceballos, E.; Álvarez-Gómez, F.; Figueroa, F.L. UVA and UVB Photoprotective capabilities of topical formulations containing mycosporine-like amino acids (MAAs) through different biological Response 11’: Necessary changes have been made in the manuscript according to the comments PA (pyruvic acid) has been defined in Page 4 Line 142 Porphyrine has been changed to porphine in Table 2 in Page 14 Lambda 1 and lambda 2 has been defined in Equation 2 Page 7 In Lines 265 and 267 it has been replaced as singlet oxygen quantum yield Figure 8 Units has been modified. |

Reviewer 2 Report
Comments and Suggestions for Authors
The manuscript includes an interesting study focused on obtaining natural sunscreens.
It is well presented and justified. My main concern is the experimental design and the corresponding statistical analysis of data.
Concrete comments would be as follows:
Abstract
Lines 10-11 include too general information.
Contrary, this section includes very scarce information on the results obtained.
Keywords
Include: aquatic matrices.
Introduction
The current study is not presented at the end of this section.
Material and methods
No indication is provided concerning the statistical analysis of data. Were there replicates carried out ? It ought to be indicated in this section and also in each Table/Figure in the Results and discussion section. As previously mentioned, this is my main concern.
In some cases, average values and standard deviations are mentioned. Does it concern replicates or several determinations in a single samples?
Comments on the Quality of English Language
Some minor editing performances could be done.
Author Response
The manuscript includes an interesting study focused on obtaining natural sunscreens. It is well presented and justified. My main concern is the experimental design and the corresponding statistical analysis of data.
Concrete comments would be as follows:
Comments 1: Abstract
Lines 10-11 include too general information. Contrary, this section includes very scarce information on the results obtained.
Response 1: Many thanks for your suggestion. We have changed the abstract as suggested in the revised manuscript.
Comments 2: Keywords
Include: aquatic matrices.
Response 2: We have included it. Thanks
Comments 3: Introduction
The current study is not presented at the end of this section.
Response 3: Many thanks for your comment. We have changed the added a final sentence describing the objective of the present work as suggested.
Comments 4: Material and methods
No indication is provided concerning the statistical analysis of data. Were there replicates carried out? It ought to be indicated in this section and also in each Table/Figure in the Results and discussion section. As previously mentioned, this is my main concern. In some cases, average values and standard deviations are mentioned. Does it concern replicates or several determinations in a single sample?
Response 4: We have added the information in the revised text.
Comments 5: Comments on the Quality of English Language
Some minor editing performances could be done.
Response 5: We have revised the text.

Reviewer 3 Report
Comments and Suggestions for Authors
In this manuscript, the authors reported findings about the photodecomposition/photostability of gadusolate and mycosporine-serinol in different water matrices. Some interesting findings have been presented. Nonetheless, some minor amendments are still required, which I have listed below. (Importantly, don’t forget to check for Figure 5. I think it is missing. There should be a DISCUSSION section too.)
1. A check on the language (grammar) and citation format of the manuscript is required.
(i) Some sentences sound incomplete/ungrammatical. For example,
lines 41-45: “Due to wide spread use of synthetic sunscreens… causing neurotoxicity, mortality in fish, reduce coral reproduction and coral reef bleaching [12].”
lines 286-287: “However, the variation in the dark … meaning that a different degradation pathway.”
(ii) Please standardize -ize vs -ise spellings. For example, both “photosensitizers” (e.g., 49 & 107) and “photosensitisers” (e.g., 175 & 240) can be found in the manuscript.
(iii) Please check and rectify incorrect in-text citation format in the manuscript. For example,
Line 30: Please check whether it should be “[1-7]” instead?
Line 37: Positions of ref #7 and ref #8 should be reversed.
Line 53: Positions of ref #15 and ref #7 should be reversed.
Line 216: “According to the literature, [39] …” should be “According to the literature [39], …”
(iv) Sometimes, required punctuation marks (periods) are missing, e.g., see lines 78 and 294.
(v) Please check for typo errors, e.g., on Line 185: “ant” should be “and”.
2. ABSTRACT
(i) This should be rewritten to make it more informative and quantitative in terms of the findings described. Highlight key data/findings quantitatively.
(ii) Results gathered should be written in the past tense, e.g., see line 16.
(iii) Lines 17-18: “The effect of … was also evaluated.” – This statement would be best replaced with a description of the findings, instead of reporting what was done.
(iv) Line 12 “mycosporine serinol” – please standardize - whether to make this term hyphenated or non-hyphenated throughout ABSTRACT and the rest of the main text.
(v) After reading the findings in ABSTRACT, it is unclear how it can be concluded that “… mycosporines could be potential candidates to be used as UV-B filters in sunscreen…”. The authors could improve this by adding at least a brief explanation/statement to before the concluding statement.
3. KEYWORDS: It is suggested that words chosen to be the ones relevant to the manuscript, but not those already appearing in the title. This may increase the discoverability of the paper (if published), when readers search literature databases, e.g., Scopus. Examples: photodegradation, photosensitizer, etc.
4. INTRODUCTION:
(i) Line 38: A few examples of “conventional synthetic sunscreens” could be provided to make the statement clearer.
(ii) Line 55: Caption of Figure 1 – It should be “structures”. The “o” in “mycosporine-serinol” needs not be in bold, correct?
(iii) Line 57: A few examples of “organic matter” could be provided to make the statement clearer.
(iv) Lines 57-59: A suitable reference should be cited to support the statement “The organic matter present … natural or man-made.”
5. MATERIALS AND METHODS
(i) Some water samples were collected from natural environments. How could the authors ensure the reproducibility of findings if the same studies were repeated in future?
(ii) The durations of the various irradiation and dark treatments have been indicated. However, it is unclear how the authors arrived at those durations? Was any preliminary testing done to determine the optimum duration of irradiation?
(iii) It would be desirable to add in a brief section in M&M detailing the statistical analysis. Information such as the software used, types of statistical tests performed, the number of replications, whether data are presented as mean ± standard errors/deviation, and the threshold for determining significance would be particularly helpful.
(iv) To avoid confusion and ensure readability, it would be desirable to not go back and forth between the full term and its abbreviation. In short, after an abbreviation has been introduced, please use it consistently. For example, after “RF” has been introduced as abbreviation for “riboflavin”, the authors used “riboflavin” again in lines 111, 113 & 162. The same goes for “porphine” and “PPY”. Please see lines 112, 113 & 162. Please check for this issue in the rest of the manuscript too.
(v) Line 90: Only “Lichina pygmaea” should be italicized.
(vi) Line 135: “PA” should be introduced in full the first time it is mentioned.
(vii) Lines 178-179: “Microbial activity was evident at night when photochemical degradation could be ruled out because of dark.” – This part is unclear. Can the authors elaborate on this? How was the observation made? Was it just based on sample cloudiness, or did it involve some measurements?
(viii) Lines 192-193: The cited refs [13] and [14] seem misplaced and shouldn’t appear before the units. Please check.
6. RESULTS
(i) The manuscript currently presents the findings in a RESULTS section WITHOUT a separate Discussion section. Did the authors intend this section to be RESULTS AND DISCUSSION instead?
(ii) It is recommended to use color lines in figures to make them clearer and easier to read, especially Figures 2, 6, 7, 8 and 12.
(iii) It is desirable to indicate number of replicates in the figure captions and include standard error/deviation bars in the line graphs to more clearly reflect variations in the dataset.
(iv) Where there are multiple charts in a figure, please indicate them as “A’, “B”, etc. For example, see Figures 3, 6, 7, 8 and 12.
(v) Figure 5 – Is this figure missing? I can see the caption (line 273). But there is nothing above the figure.
(vi) Figure 6 – The data for 2 hours are presented for the first chart (estuary), but not for the other three charts. Why? Please check.
(vii) Figure 7 –
The data for 0 hours are not presented for the first chart (pure water-Gd). Why? Please check.
The data for 3 hours, instead of for 4 hours, are presented for the fourth chart (ocean water-M-ser(OH)), unlike the other three charts. Why? Please check.
(viii) Figure 8 – Please correct spelling mistake in y axis title (“Normelised”) of both charts.
(ix) Figures 9 and 13 –
It would be less confusing to standardize the term, instead of using “photodegradation” in y-axis title, then switching to “photodecomposition” in figure caption.
In addition, to facilitate data interpretation and enable an objective comparison (e.g., on lines 356-357), please consider conducting a statistical analysis of the bar chart data. A multiple comparison of means, such as Tukey’s test, would be appropriate.
Please also indicate the number of replicates (n) for each group.
Please also clarify in the caption what the vertical lines represent: are they standard errors (SE) or standard deviations (SD)?
(x) Figure 10 – Caption can be improved. Please indicate the meaning of the different symbols, what the vertical lines represent (SE/SD?) and the number of replicates.
(xi) Figure 11 – Caption can be improved. Please indicate meaning of “Sens”, “1Sens*” and “3Sens*”.
(xii) Figure 12 –
Font size can be increased.
Name each chart as “A”, “B”…
Using color lines will make it easier to read. In the current version, it is difficult/impossible to distinguish the lines representing different irradiation times.
Please indicate what the vertical lines represent (SE/SD?) and the number of replicates.
(xiii) Table 1 –
Statistical analysis is required to for the data presented.
Please indicate what the data are mean ± SE/SD and the number of replicates.
The text immediately below the table – it looks like footnote but also like the main text (a new paragraph). Which is it?
(xiv) Table 2 –
It would be less confusing to standardize the term, instead of using “Photodegradation quantum yield” in table caption, and then switching to “Quantum Yield of Photodegradation” in table heading.
Is it necessary to show “0.6 ± 0.4” in bold?
Statistical analysis is required to for the data presented.
Please indicate what the data are mean ± SE/SD and the number of replicates.
The text immediately below the table – it looks like footnote but also like the main text (a new paragraph). Which is it?
(xv) Lines 247-248 compared the data for river, estuary and ocean water as shown in the table. My concern is that, as shown in the table, the irradiation times differed between the samples. So, in this case, it is still reasonable to compare them directly? I have the same concern about the comparison between pure and natural water samples on lines 263-264 too. It would be desirable to clarify/justify this in the text.
(xvi) Line 297 – It should be Figure 7, not Figure 3, correct?
(xvii) Line 308 – After the mention of Table 2, the table should be shown as immediately as possible, not after another 5-6 paragraphs and 4 figures.
(xviii) Line 320 –
“(0.2±0.4)” – The variation in the data seems very large – is this a typo error?
“Gd” – The authors appeared to randomly switch between abbreviation “Gd” and the full term “gadusolate” in the main text. Please be consistent.
(xix) Line 332 – “… (1.6±0.2)×10-4 probably…” – some info is missing before the word “probably”.
(xx) Line 334 – Does “… 4h30…” refer to 4.5 hours?
(xxi) Lines 335-336 – Can “(0.8±0.2)×10-4” be considered “in agreement” with “(1.27±0.32)×10-4”? The latter is about 60% higher in value versus the former. Some elaboration/revision seems required here.
(xxii) Lines 378-379 – Please cite suitable references to support the statement that “These reduction potential values make it thermodynamically favourable.”
- CONCLUSION
(i) Is there any distinct advantage/potential difference between Gd and M-ser(OH) as UV filters based on the results gathered?
(ii) The connection between the findings (lines 383-387) and the conclusion that the molecules could be potential candidates for UV-B filters in sunscreen formulations (lines 387-388) is not explicitly clear. To improve the flow of ideas, the authors could include a brief explanatory statement linking the findings to this conclusion. Alternatively, the last concluding sentence could also be revised to focus more on addressing the objective of the study instead.
Comments on the Quality of English Language
A check on the language (grammar) is required.
Author Response
|
In this manuscript, the authors reported findings about the photodecomposition/photostability of gadusolate and mycosporine-serinol in different water matrices. Some interesting findings have been presented. Nonetheless, some minor amendments are still required, which I have listed below. (Importantly, don’t forget to check for Figure 5. I think it is missing. There should be a DISCUSSION section too.) Response 0: Thanks for your comments, we have added Fig 5 and ‘Discussion’. We would like to thanks this referee for the suggestions and comments on the manuscript. The comments greatly improved the quality of the article. Thanks. 1. A check on the language (grammar) and citation format of the manuscript is required. Comments 1: Some sentences sound incomplete/ungrammatical. For example, lines 41-45: “Due to wide spread use of synthetic sunscreens… causing neurotoxicity, mortality in fish, reduce coral reproduction and coral reef bleaching [12].” lines 286-287: “However, the variation in the dark … meaning that a different degradation pathway.” Response 1: The lines 45-49 in page 1, have been modified as stated by the authors as Due to wide spread use of synthetic sunscreens along with the recreational activities, these compounds ending up in aquatic ecosystems like ocean, rivers and estuaries and also inside marine organisms such as fishes, coral reefs, etc. causing neurotoxicity, mortality in fish, reduce coral reproduction and coral reef bleaching. The sentence has been modified as stated by the authors However, the degradation in the dark is low relative to the irradiated molecules indicating that biotic degradation is relatively slow. |
|
Comments 2: Please standardize -ize vs -ise spellings. For example, both “photosensitizers” (e.g., 49 & 107) and “photosensitisers” (e.g., 175 & 240) can be found in the manuscript. |
|
Response 2: The spelling has been modified accordingly as ‘photosensitisers’ in the entire revised manuscript. Comments 3: Please check and rectify incorrect in-text citation format in the manuscript. For example,Line 30: Please check whether it should be “[1-7]” instead? Line 37: Positions of ref #7 and ref #8 should be reversed. Line 53: Positions of ref #15 and ref #7 should be reversed. Line 216: “According to the literature, [39] …” should be “According to the literature [39], …” Response 3: The comments of the referee has been taken into account in the revised manuscript; references are inter changed in line 39 (ref 7, ref 10) and line 55 (ref 7, ref 16). It was modified according to the literature [41]. Comments 4: Sometimes, required punctuation marks (periods) are missing, e.g., see lines 78 and 294. Response 4: The comments of the referee has been taken into account and the manuscript has been changed. Comments 5: Please check for typo errors, e.g., on Line 185: “ant” should be “and”. Response 5: The comments have been taken into account and manuscript was changed accordingly. Abstract Comments 6: i) This should be rewritten to make it more informative and quantitative in terms of the findings described. Highlight key data/findings quantitatively (ii) Results gathered should be written in the past tense, e.g., see line 1 (iii) Lines 17-18: “The effect of … was also evaluated.” – This statement would be best replaced with a description of the findings, instead of reporting what was done (iv) Line 12 “mycosporine serinol” – please standardize - whether to make this term hyphenated or non-hyphenated throughout ABSTRACT and the rest of the main text.(v) After reading the findings in ABSTRACT, it is unclear how it can be concluded that “… mycosporines could be potential candidates to be used as UV-B filters in sunscreen…”. The authors could improve this by adding at least a brief explanation/statement to before the concluding statement. Response 6: These changes have been taken into account in the revised manuscript; line 16 in page 1 has been ‘The study also revealed that the photodegradation of these mycosporines were higher in these natural matrices than in pure water due to the generation of singlet oxygen on UV irradiation’. Line 17-19 in page 1: ‘These could be potential candidates to be used as UV-B filters in sunscreen formulations due to its biobased, ecofriendly nature, high photostability and inherent biodegradability, in contrary to the current existing conventional sunscreens.’ Comments 7: 3. KEYWORDS: It is suggested that words chosen to be the ones relevant to the manuscript, but not those already appearing in the title. This may increase the discoverability of the paper (if published), when readers search literature databases, e.g., Scopus. Examples: photodegradation, photosensitizer, etc. Response 7: The keywords ‘photodegradation, photosensitiser’ were added. Comments 8: INTRODUCTION: (i) Line 38: A few examples of “conventional synthetic sunscreens” could be provided to make the statement clearer.(ii) Line 55: Caption of Figure 1 – It should be “structures”. The “o” in “mycosporine-serinol” needs not be in bold, correct? (iii) Line 57: A few examples of “organic matter” could be provided to make the statement clearer. (iv) Lines 57-59: A suitable reference should be cited to support the statement “The organic matter present … natural or man-made.” Response 8: Line 40-41 page 1: Because of some controversy related to side effects provoked by the conventional synthetic sunscreens e.g., TiO2, ZnO etc. Line 58 page 1: Chemical structures of: gadusol, gadusolate and mycosporine-serinol (M-ser(OH)) Line 60-62 page 2: The organic matter (e.g., humic acid, fulvic acid) present in natural water matrices, like river, estuary and ocean, is known to generate ROS upon solar irradiation and thus helping to transform organic molecules, whether natural or man-made. Comments 9 : MATERIALS AND METHODS (i) Some water samples were collected from natural environments. How could the authors ensure the reproducibility of findings if the same studies were repeated in future? (ii) The durations of the various irradiation and dark treatments have been indicated. However, it is unclear how the authors arrived at those durations? Was any preliminary testing done to determine the optimum duration of irradiation? (iii) It would be desirable to add in a brief section in M&M detailing the statistical analysis. Information such as the software used, types of statistical tests performed, the number of replications, whether data are presented as mean ± standard errors/deviation, and the threshold for determining significance would be particularly helpful. (iv) To avoid confusion and ensure readability, it would be desirable to not go back and forth between the full term and its abbreviation. In short, after an abbreviation has been introduced, please use it consistently. For example, after “RF” has been introduced as abbreviation for “riboflavin”, the authors used “riboflavin” again in lines 111, 113 & 162. The same goes for “porphine” and “PPY”. Please see lines 112, 113 & 162. Please check for this issue in the rest of the manuscript too. (v) Line 90: Only “Lichina pygmaea” should be italicized. (vi) Line 135: “PA” should be introduced in full the first time it is mentioned. (vii) Lines 178-179: “Microbial activity was evident at night when photochemical degradation could be ruled out because of dark.” – This part is unclear. Can the authors elaborate on this? How was the observation made? Was it just based on sample cloudiness, or did it involve some measurement? (viii) Lines 192-193: The cited refs [13] and [14] seem misplaced and shouldn’t appear before the units. Please check. Response 9: (i) The water samples from the natural environment were collected from the same source at different intervals and the experiments were repeated and reproducible results were obtained. (ii) These durations were optimized via trial-and-error methods. (iii) The number of replications was added in the revised manuscript and the data are presented as mean ± standard errors/deviation. We used Excel to treat our results. (iv) Riboflavin and porphine have been used in the entire manuscript and abbreviations RF and PPY are removed. (v) Now, only Lichina pygmaea was italicized. (vi) The full form of PA has been mentioned. (vii) We remove the sentence, in order to not provoke any confusion. The number of replicates was added. (viii) The references were checked and changed. Comments 10: RESULTS (i) The manuscript currently presents the findings in a RESULTS section WITHOUT a separate Discussion section. Did the authors intend this section to be RESULTS AND DISCUSSION instead? (ii) It is recommended to use color lines in figures to make them clearer and easier to read, especially Figures 2, 6, 7, 8 and 12. (iii) It is desirable to indicate number of replicates in the figure captions and include standard error/deviation bars in the line graphs to more clearly reflect variations in the dataset. (iv) Where there are multiple charts in a figure, please indicate them as “A’, “B”, etc. For example, see Figures 3, 6, 7, 8 and 12. (v) Figure 5 – Is this figure missing? I can see the caption (line 273). But there is nothing above the figure. (vi) Figure 6 – The data for 2 hours are presented for the first chart (estuary), but not for the other three charts. Why? Please check. (vii) Figure 7 – The data for 0 hours are not presented for the first chart (pure water-Gd). Why? Please check. The data for 3 hours, instead of for 4 hours, are presented for the fourth chart (ocean water-M-ser(OH)), unlike the other three charts. Why? Please check (viii) Figure 8 – Please correct spelling mistake in y axis title (“Normelised”) of both charts. (ix) Figures 9 and 13 – It would be less confusing to standardize the term, instead of using “photodegradation” in y-axis title, then switching to “photodecomposition” in figure caption. In addition, to facilitate data interpretation and enable an objective comparison (e.g., on lines 356-357), please consider conducting a statistical analysis of the bar chart data. A multiple comparison of means, such as Tukey’s test, would be appropriate. Please also indicate the number of replicates (n) for each group. Please also clarify in the caption what the vertical lines represent: are they standard errors (SE) or standard deviations (SD)? (x) Figure 10 – Caption can be improved. Please indicate the meaning of the different symbols, what the vertical lines represent (SE/SD?) and the number of replicate (xi) Figure 11 – Caption can be improved. Please indicate meaning of “Sens”, “1Sens*” and “3Sens*”. (xii) Figure 12 – ----Font size can be increased. ---Name each chart as “A”, “B”… ---Using color lines will make it easier to read. In the current version, it is difficult/impossible to distinguish the lines representing different irradiation times. ---Please indicate what the vertical lines represent (SE/SD?) and the number of replicates. (xiii) Table 1 – ---Statistical analysis is required to for the data presented. ----Please indicate what the data are mean ± SE/SD and the number of replicates. ---The text immediately below the table – it looks like footnote but also like the main text (a new paragraph). Which is it? (xiv) Table 2 – - --It would be less confusing to standardize the term, instead of using “Photodegradation quantum yield” in table caption, and then switching to “Quantum Yield of Photodegradation” in table heading. ----Is it necessary to show “0.6 ± 0.4” in bold? --Statistical analysis is required to for the data presented. ---Please indicate what the data are mean ± SE/SD and the number of replicates. ---The text immediately below the table – it looks like footnote but also like the main text (a new paragraph). Which is it? (xv) Lines 247-248 compared the data for river, estuary and ocean water as shown in the table. My concern is that, as shown in the table, the irradiation times differed between the samples. So, in this case, it is still reasonable to compare them directly? I have the same concern about the comparison between pure and natural water samples on lines 263-264 too. It would be desirable to clarify/justify this in the text. (xvi) Line 297 – It should be Figure 7, not Figure 3, correct? (xvii) Line 308 – After the mention of Table 2, the table should be shown as immediately as possible, not after another 5-6 paragraphs and 4 figures. (xviii) Line 320 – (0.2±0.4)” – The variation in the data seems very large – is this a typo error? · “Gd” – The authors appeared to randomly switch between abbreviation “Gd” and the full term “gadusolate” in the main text. Please be consistent. (xix) Line 332 – “… (1.6±0.2)×10-4 probably…” – some info is missing before the word “probably”. (xx) Line 334 – Does “… 4h30…” refer to 4.5 hours? (xxi) Lines 335-336 – Can “(0.8±0.2)×10-4” be considered “in agreement” with “(1.27±0.32)×10-4”? The latter is about 60% higher in value versus the former. Some elaboration/revision seems required here. (xxii) Lines 378-379 – Please cite suitable references to support the statement that “These reduction potential values make it thermodynamically favourable.” Response 10 : (i) Thanks for your comment. We added DISCUSSION in the RESULTS and DISCUSSION section. (ii) We thank the suggestion of the referee. We tried to do it, but in our opinion, our chose using single color or bicolor with dots and full lines is clearer and easier to read. (iii) The revised manuscript was modified - 3 replicates are added in the Figure captions for all figures. The graphs were modified as suggested. (iv) Corrections were made in the manuscript a), and b) has been indicated in the figures with multiple charts. (v) We have added Figure 5. Thanks. (vi) We have changed the figure as suggested. (vii) We have changed the figure as suggested. (viii) We have changed the figure as suggested. (ix) We have changed the figure as suggested. (x) We have changed the figure as suggested. (xi) We have changed the figure as suggested. (xiii) We have changed the figure as suggested. (xiv) Correction has been made in the entire manuscript. It was standardized as photodegradation quantum yield. (xv) The irradiation times are optimized according to the organic matter present in the natural matrices. River water has the highest organic matter so it was just irradiated for 1 hr , river and estuary water has low organic matter so they were irradiated for 4 h. The quantum yield of singlet oxygen generated by photosensitisers is on the higher side compared to quantum yield of singlet oxygen in natural matrices generated from organic matter. (xvi) Corrections has been made Figure 7. (xvii) We have rearranged the Table and Figures. (xviii) Line 323 in page 10 was corrected as (0.2 ± 00.4). It was an error. In the entire manuscript Gd was replaced by gadusolate (except for representing concentration, quantum yield and radicals it was denoted as Gd). (xix) We have modified as: It is slightly higher in artificial sea (2.7±0.6)×10-4 for gadusolate and M-ser(OH)(1.6±0.2)×10-4 probably due to the pH increase from 7 to 8.2. (xix) Line 337 page 10 was modified as 4.5 h. (xxi) We are sorry, but we did not understand the referee’ comment. (xxii) The redox potential of gadusol, triplet state of riboflavin and porphine are 0.60 V, 1.7 V and 1.54 V respectively. As a result the type 1 photosensitisation is said to be thermodynamically favorable. Attached Equation below Comments 11 7. CONCLUSION (i) Is there any distinct advantage/potential difference between Gd and M-ser(OH) as UV filters based on the results gathered? (ii) The connection between the findings (lines 383-387) and the conclusion that the molecules could be potential candidates for UV-B filters in sunscreen formulations (lines 387-388) is not explicitly clear. To improve the flow of ideas, the authors could include a brief explanatory statement linking the findings to this conclusion. Alternatively, the last concluding sentence could also be revised to focus more on addressing the objective of the study instead. Response 11 (i) Our results clearly indicate that there is a difference in stability between the two compounds studied. We have changed the wording in the conclusion to state that ‘This study infers that M-ser(OH) was more photostable relative to gadusolate probably due to the presence of the bulky (1,3-dihydroxypropan-2-yl) moiety’. (ii) The last sentences were modified in the manuscript accordingly. |

Reviewer 4 Report
Comments and Suggestions for Authors
The study compared two UV absorbing mycosporines and found one is deserved practical application. The conclusion is supported by sufficient data and all the sections are well organized with clear and logical description.
minor comment:
(1) in abstract. quantitative data of key results should be added here to support the main conclusions.
(2) in results and discussion part. comparison and discussion of the performance (especially stability) of other mycosporines is lacked.
Author Response
|
The study compared two UV absorbing mycosporines and found one is deserved practical application. The conclusion is supported by sufficient data and all the sections are well organized with clear and logical description. minor comment: Comments 1: In abstract. quantitative data of key results should be added here to support the main conclusions. |
|
Response 1: The Manuscript has been revised taking into account the suugestions of the referee. Line 14-16 page 1 In terms of photostability, both gadusol and mycosporine-serinol filters were found to offer good protection and high performance in terms of photodegadation quantum yield (0.8 ± 0.2)10-4 and (1.1 ± 0.6)10-4, respectively |
|
Comments 2: In results and discussion part. comparison and discussion of the performance (especially stability) of other mycosporines is lacked. |
|
Response 2: This general suggestion of the referee, was taking in account in the revised manuscript at the same time of other suggestions of the other referees. Many thanks for your suggestions.
|

Round 2
Reviewer 1 Report
Comments and Suggestions for Authors
The manuscript has been improved and most of the questions previously raised have been answered but still, some issues have not been considered carefully, or solved satisfactorily enough. Particularly, the discussion of the results needs important refinement. On this basis, I recommend accepting the paper after the modifications I suggest below are incorporated and more carefully edition is achieved.
Regarding Comments 1 of the 1st report:
The text still mentions the two studied compounds (mycosporine serinol and gadusolate) as “the mycosporines” or “the two mycosporines”. See for instance lines 195, 334, 344, 368, 373, and 384. Thus, the manuscript must be completely revised again to solve the inconsistencies in the designation of the molecules focused in this research.
About Comments 2:
The new version of the introduction part, lines 41-42, says “ ..the conventional synthetic sunscreens e.g., TiO2, ZnO, etc….”. This is mistaken since TiO2 and ZnO are not synthetic substances but minerals (i.e. found in Nature). The authors should revise these concepts in the papers cited here and in the references therein.
About Comments 3:
I think that the modifications made in the abstract, introduction, and conclusion sections to account for the evaluation of the performance of the natural metabolites in photoprotection are still not convincing. The authors insist that these molecules “…could be potential candidates to be used as UV-B filters in sunscreen formulations due to its biobased and eco-friendly nature, inherent biodegradability …” although this study is not directed to prove particularly any of these properties (biodegradability, eco-friendly nature, or biological origin) but instead to assess the photostability in natural water matrices and the effect of ROS. Thus, the paper must precisely explain why the behavior of the two metabolites in the various aqueous environments and the effect of photosensitized ROS are both relevant aspects to support their potential use as UV-B filters.
About Comments 4:
The new version of the manuscript allows us to see figure 5 so that the issues that were expressed in this comment are now clear. The manuscript may benefit by remarking in the experimental part the relative (minor) intensity of the spectral distribution of the radiation source in the interval 250-290 nm, concerning achieving solar simulation.
Besides, the authors answer that “All the spectra have been normalized accordingly”. By which procedure exactly? This must be precisely specified in the text.
About Comments 5:
Analogously, the filtering treatment applied on the water samples to make the scattering parameter negligible must be reported in the experimental part (type filter used: material, pore size)
About Comments 7:
I understand that quantum yields of photodegradation are independent of the initial concentrations, but precisely the text refers to Fig. 7 to conclude about the reactivities. No information on quantum yields can be derived only from these plots, just degradation velocities. The authors should consider that the parameter “quantum yield” relates the reaction rates with the photon absorption rates, which in turn do depend on the initial concentration and molar absorption coefficients of the reactants. In this sense, I suggest revising the text and directing the discussion to compare the quantum yields of the two compounds instead of absorption changes alone.
About Comments 8:
If I am not mistaken, by comparing the photodegradation quantum yields in the presence of azide for the two photosensitizers (Table 2), riboflavin´s values are around 3 or 5 times larger than porphine´s. Since the azides quench singlet oxygen but not superoxide anion radical, could you discard the role of this last species on the photodecomposition of mycosporine serine or gadusolate?
About Comments 9:
I agree that the work is not aimed at a photophysical study. That is why I observed that your conclusion about the role of triplet sensitizers in photodecomposition is too definite. More experiments are needed to support such a conclusive statement. I recommend re-writing this sentence (lines 386-387) so that it suggests that the result is just in line with a certain mechanism but it is not able to confirm it.
About Comments 10:
The comparison of the photodecomposition quantum yields in pure water is strongly contradictory. In the abstract (lines 13-16) the respective results are mentioned (0.8 ± 0.2)10-4 and (1.1 ± 0.6)10-4, which clearly show that the yields are the same within the experimental error (and so it is concluded in lines 351-356). However, in the following lines in the abstract, it is affirmed that “.. The photostability of mycosporine-serinol was found to be superior to that of gadusolate for all water matrices..,” which is not true according to both the numbers given before for pure water and the results reported in Table 2 (particularly the case of ocean water). Then, in the conclusions part it is again said that mycosporine serine is more photostable than gadusolate (lines 409-410).
In the response, the authors express that the results “clearly indicate that there is a difference in stability between the two compounds studied”, which is far from evident, as commented above. Thus, this serious flaw of the analysis must be revised carefully and discussed/re-written consistently.
Besides, a possible connection between the bulkiness of substituent moieties and the photostability of a molecule is not obvious at all. Thus, any difference observed in the photoreactivity that is proposed to be justified in terms of the molecular structure must be unavoidably supported by references to previous photochemical-photophysical studies indicating some trend or effect of these features on the relaxation mechanisms of the excited electronic states.
About Comments 11:
Again, Arbeloa´s paper does not correspond to ref.#16 (line 359). This was not corrected in the new version of the manuscript.
Other suggestions or revisions required:
-Lines 28-32: avoid repetition of the verb “possess”
-Table 2. Please consider replacing “Raw materials” with more appropriate words. For instance: (natural) compound, substance…; and “artificial seawater” instead of “artificial sea”
-Correct Ref. # 26 (Bens, G. Sunscreens. In: Sunlight, Vitamin D and Skin Cancer. Springer, New York, NY. 2014, 429-463 ) It does not contain any report on the extraction or purification of M-ser(OH) (see line 100)
-Please define “lambda max” and “lambda min” for equation 2 (line 193)
-Equation 4, line 192. Keep the nomenclature to indicate the numbers of photons as used for equation 3 (N?) instead of writing “mol”. Eliminate parenthesis.
-Line 280: write “the number of photons in mol” or “the number of photon mols”
-Make uniform throughout the manuscript the use of capital letters or not when writing Gadusol, Gadusolate, and Mycosporine.
-Line 292: It is said that “ the M-ser(OH) absorbance spectra remain unchanged under the same conditions (Figure 6)…”. However, Fig. 6 c) shows changes in time of the spectra for estuary water. Please, revise the text and correct it according to the results.
-Line 305: Omit “The” in “…The gadusol with a pKa at 4.25 was…”
-Line 368: “Sensitized photodegradation pathways…”
-Page 12, Figure 12. It is hard to read the insets, increase font size. In a) time unit in the inset is missed.
-Line 379: Include decimal figures: 4.0 instead of 4
-Lines 382-383: “…sodium azide that quenches singlet oxygen in…”
-Line 396: “the carbonyl group of the cyclohexenone residue was proposed”... On which basis? It must be specified or accordingly referred.
-Lines 398-399: About the statement on gadusolate reactions with triplet sensitizers: is this suggested by the authors? O was it previosly determined? As far as I can check, this is not particularly reported in ref. # 52, 53 nor 54.
-Lines 403-404: “The redox potentials of …… were found to be …”
Comments on the Quality of English Language
The writing must be reviewed mainly for consistency in verb conjugation and repetition of vocabulary.
Author Response
|
The manuscript has been improved and most of the questions previously raised have been answered but still, some issues have not been considered carefully, or solved satisfactorily enough. Particularly, the discussion of the results needs important refinement. On this basis, I recommend accepting the paper after the modifications I suggest below are incorporated and more carefully edition is achieved. Comment 1: Regarding Comments 1 of the 1st report: The text still mentions the two studied compounds (mycosporine serinol and gadusolate) as “the mycosporines” or “the two mycosporines”. See for instance lines 195, 334, 344, 368, 373, and 384. Thus, the manuscript must be completely revised again to solve the inconsistencies in the designation of the molecules focused in this research. Response 1: Many thanks for your remark. The manuscript was completely revised to solve the inconsistencies. Modifications are done in blue. Comment 2: About Comments 2: The new version of the introduction part, lines 41-42, says “ ..the conventional synthetic sunscreens e.g., TiO2, ZnO, etc….”. This is mistaken since TiO2 and ZnO are not synthetic substances but minerals (i.e. found in Nature). The authors should revise these concepts in the papers cited here and in the references therein. Response 2: Many thanks for your remark. We have changed the sentence in the revised manuscript: ‘Because of some controversy related to side effects provoked by the conventional synthetic sunscreens and mineral sunscreens (e.g., TiO2, ZnO, etc.) on both human and environment health…’ Comment 3: About Comments 3: I think that the modifications made in the abstract, introduction, and conclusion sections to account for the evaluation of the performance of the natural metabolites in photoprotection are still not convincing. The authors insist that these molecules “…could be potential candidates to be used as UV-B filters in sunscreen formulations due to its biobased and eco-friendly nature, inherent biodegradability …” although this study is not directed to prove particularly any of these properties (biodegradability, eco-friendly nature, or biological origin) but instead to assess the photostability in natural water matrices and the effect of ROS. Thus, the paper must precisely explain why the behavior of the two metabolites in the various aqueous environments and the effect of photosensitized ROS are both relevant aspects to support their potential use as UV-B filters. Response 3: The reviewer is absolutely right. Thanks for your comment. We changed the abstract and the conclusion. Please see the revised manuscript in blue. Comment 4: About Comments 4: The new version of the manuscript allows us to see figure 5 so that the issues that were expressed in this comment are now clear. The manuscript may benefit by remarking in the experimental part the relative (minor) intensity of the spectral distribution of the radiation source in the interval 250-290 nm, concerning achieving solar simulation. Besides, the authors answer that “All the spectra have been normalized accordingly”. By which procedure exactly? This must be precisely specified in the text. Response 4: We are sorry for this mistake in response to comment 4 regarding round 1. The spectra of Fig. 5 were not normalized. The axes of the figure show the SI units. Comment 5: About Comments 5: Analogously, the filtering treatment applied on the water samples to make the scattering parameter negligible must be reported in the experimental part (type filter used: material, pore size) Response 5: Thanks for your suggestion. This information was added in the revised manuscript (line 104-105) Comment 7: About Comments 7: I understand that quantum yields of photodegradation are independent of the initial concentrations, but precisely the text refers to Fig. 7 to conclude about the reactivities. No information on quantum yields can be derived only from these plots, just degradation velocities. The authors should consider that the parameter “quantum yield” relates the reaction rates with the photon absorption rates, which in turn do depend on the initial concentration and molar absorption coefficients of the reactants. In this sense, I suggest revising the text and directing the discussion to compare the quantum yields of the two compounds instead of absorption changes alone. Response 7: Thanks for your comment. The presence of Fig. 7 in the manuscript was to show that we used the variation of the absorbance to illustrate the degradation under solar irradiation. We did not discuss absorbance in the text. Please, see line 341-367. Comment 8: About Comments 8: If I am not mistaken, by comparing the photodegradation quantum yields in the presence of azide for the two photosensitizers (Table 2), riboflavin´s values are around 3 or 5 times larger than porphine´s. Since the azides quench singlet oxygen but not superoxide anion radical, could you discard the role of this last species on the photodecomposition of mycosporine serine or gadusolate? Response 8: Thanks for your comment. We cannot discard the role of other species on the photodegradation. Comment 9: About Comments 9: I agree that the work is not aimed at a photophysical study. That is why I observed that your conclusion about the role of triplet sensitizers in photodecomposition is too definite. More experiments are needed to support such a conclusive statement. I recommend re-writing this sentence (lines 386-387) so that it suggests that the result is just in line with a certain mechanism but it is not able to confirm it. Response 9: The sentence was modified as suggested in the revised manuscript. Comment 10: About Comments 10: The comparison of the photodecomposition quantum yields in pure water is strongly contradictory. In the abstract (lines 13-16) the respective results are mentioned (0.8 ± 0.2)10-4 and (1.1 ± 0.6)10-4, which clearly show that the yields are the same within the experimental error (and so it is concluded in lines 351-356). However, in the following lines in the abstract, it is affirmed that “.. The photostability of mycosporine-serinol was found to be superior to that of gadusolate for all water matrices..,” which is not true according to both the numbers given before for pure water and the results reported in Table 2 (particularly the case of ocean water). Then, in the conclusions part it is again said that mycosporine serine is more photostable than gadusolate(lines 409-410). Response 10: Thanks for the remark, we have changed the abstract and conclusion of the revised manuscript. Comment 10’: In the response, the authors express that the results “clearly indicate that there is a difference in stability between the two compounds studied”, which is far from evident, as commented above. Thus, this serious flaw of the analysis must be revised carefully and discussed/re-written consistently. Response 10’: This was done in the new revision. Comment 10’’: Besides, a possible connection between the bulkiness of substituent moieties and the photostability of a molecule is not obvious at all. Thus, any difference observed in the photoreactivity that is proposed to be justified in terms of the molecular structure must be unavoidably supported by references to previous photochemical-photophysical studies indicating some trend or effect of these features on the relaxation mechanisms of the excited electronic states. Response 10’’: Many thanks for your comments. The referee is completely right. These aspects were revised in the manuscript (line 405). Comment 11: About Comments 11: Again, Arbeloa´s paper does not correspond to ref.#16 (line 359). This was not corrected in the new version of the manuscript. Other suggestions or revisions required: -Lines 28-32: avoid repetition of the verb “possess” -Table 2. Please consider replacing “Raw materials” with more appropriate words. For instance: (natural) compound, substance…; and “artificial seawater” instead of “artificial sea” -Correct Ref. # 26 (Bens, G. Sunscreens. In: Sunlight, Vitamin D and Skin Cancer. Springer, New York, NY. 2014, 429-463 ) It does not contain any report on the extraction or purification of M-ser(OH) (see line 100) -Please define “lambda max” and “lambda min” for equation 2 (line 193) -Equation 4, line 192. Keep the nomenclature to indicate the numbers of photons as used for equation 3 (N?) instead of writing “mol”. Eliminate parenthesis. -Line 280: write “the number of photons in mol” or “the number of photon mols” -Make uniform throughout the manuscript the use of capital letters or not when writing Gadusol, Gadusolate, and Mycosporine. -Line 292: It is said that “ the M-ser(OH) absorbance spectra remain unchanged under the same conditions (Figure 6)…”. However, Fig. 6 c) shows changes in time of the spectra for estuary water. Please, revise the text and correct it according to the results. -Line 305: Omit “The” in “…The gadusol with a pKa at 4.25 was…” -Line 368: “Sensitized photodegradation pathways…” -Page 12, Figure 12. It is hard to read the insets, increase font size. In a) time unit in the inset is missed. -Line 379: Include decimal figures: 4.0 instead of 4 -Lines 382-383: “…sodium azide that quenches singlet oxygen in…” -Line 396: “the carbonyl group of the cyclohexenone residue was proposed”... On which basis? It must be specified or accordingly referred. -Lines 398-399: About the statement on gadusolate reactions with triplet sensitizers: is this suggested by the authors? O was it previosly determined? As far as I can check, this is not particularly reported in ref. # 52, 53 nor 54. -Lines 403-404: “The redox potentials of …… were found to be …” Response 11: Thanks for your help regarding all these points. - Arbeloa´s paper ref was changed for ref.#20 (line 359). -Lines 28-32: “possess” was replaced by ‘have’ -Table 2. “Raw materials” was replaced by ‘Natural compounds’ and “artificial seawater” instead of “artificial sea” -The Ref. # 26 was changed to Ref # 19 -Please define “lambda max” and “lambda min” for equation 2 (line 193). This was done in the revised manuscript. -Equation 4, line 192. N was used instead of mol. The parenthesis was eliminated. -Line 280: the chaged was done -Make uniform throughout the manuscript the use of capital letters or not when writing Gadusol, Gadusolate, and Mycosporine. It was done. -Line 292: It is said that “ the M-ser(OH) absorbance spectra remain unchanged under the same conditions (Figure 6)…”. Thanks for your remark. The text was changed. -Line 305: the chaged was made -Line 368: the changed was done. -Page 12, Figure 12. This was done in the revised manuscript. -Line 379: the changed was done. -Lines 382-383: the changed was done. -Line 396: “the carbonyl group of the cyclohexenone residue was proposed”... This was clarified in the revised manuscript. -Lines 398-399: We have suggested and also, we revised the references. -Lines 403-404: the changed was done. |
All the references were revised.
The changes were made in blue.

Reviewer 2 Report
Comments and Suggestions for Authors
The manuscript has been performed taking into account most of the previous comments. However, I feel one important aspect is still not addressed. The authors still have not indicated the statistical treatment carried out to data. In the new version, they have indicated the number of replicates. However, the manuscript includes words like increase, decrease, higher, lower, … Such statements ought to be based on a certain statistical treatment that is not expressed in the materials and methods section neither in data include in Tables and Figures.
Author Response
The manuscript has been performed taking into account most of the previous comments. However, I feel one important aspect is still not addressed.
Comments 1: The authors still have not indicated the statistical treatment carried out to data. In the new version, they have indicated the number of replicates. However, the manuscript includes words like increase, decrease, higher, lower, … Such statements ought to be based on a certain statistical treatment that is not expressed in the materials and methods section neither in data include in Tables and Figures.
Response 1: We have added the statical treatment in the experimental part.